# HybridOM: Hybrid Physics-Based and Data-Driven Global Ocean Modeling with Efficient Spatial Downscaling

**Ruiqi Shu** [1][2]   **Xiaohui Zhong** [3][2]   **Qiusheng Huang** [3][4][2]   **Ruijian Gou** [5]   **Tianrun Gao** [6][2]   **Hao Li** [3][4][2]
**Xiaomeng Huang** [1]

## Abstract

Global ocean modeling is vital for climate science but struggles to balance computational efficiency with accuracy. Traditional numerical solvers are accurate but computationally expensive, while pure deep learning approaches, though fast, often lack physical consistency and long-term stability. To address this, we introduce HybridOM, a framework integrating a lightweight, differentiable numerical solver as a skeleton to enforce physical laws, with a neural network as the flesh to correct subgrid-scale dynamics. To enable efficient high-resolution modeling, we further introduce a physics-informed regional downscaling mechanism based on flux gating. This design achieves the inference efficiency of AI-based methods while preserving the accuracy and robustness of physical models. Extensive experiments on the GLORYS12V1 and OceanBench dataset validate HybridOM's performance in two distinct regimes: long-term subseasonal-to-seasonal simulation and short-term operational forecasting coupled with the FuXi-2.0 weather model. Results demonstrate that HybridOM achieves state-of-the-art accuracy while strictly maintaining physical consistency, offering a robust solution for next-generation ocean digital twins. Our source code is available at https://github.com/ChiyodaMomo01/HybridOM.

---

[1]Department of Earth System Science, Tsinghua University, Beijing, China [2]Shanghai Academy of Artificial Intelligence for Science, Shanghai, China [3]Artificial Intelligence Innovation and Incubation Institute, Fudan University, Shanghai, China [4]Shanghai Innovation Institute, Shanghai, China [5]Department of Ocean Big Data and Artificial Intelligence, Laoshan National Laboratory, Qingdao, China [6]Department of Geotechnical Engineering, Tongji University, Shanghai, China. Correspondence to: Xiaomeng Huang <hxm@tsinghua.edu.cn>, Hao Li <li-hao_lh@fudan.edu.cn>.

*Proceedings of the 43$^{rd}$ International Conference on Machine Learning*, Seoul, South Korea. PMLR 306, 2026. Copyright 2026 by the author(s).

## 1. Introduction

Global ocean simulation is fundamental to climate science and marine ecosystem management but faces the dual challenge of chaotic multiscale dynamics and prohibitive computational costs (Cui et al., 2025; Fox-Kemper et al., 2019). While accurate subseasonal-to-seasonal (S2S) forecasting is critical for the blue economy and disaster mitigation (Hobday et al., 2016; Guo et al., 2025), resolving these complex interactions remains a daunting computational task.

Ocean forecasting has historically bifurcated into physics-based numerical models, which ensure rigorous physical consistency and interpretability (Griffies et al., 2005), and emerging deep learning emulators that achieve orders-of-magnitude speedups (Cui et al., 2025; Wang et al., 2024; Xiong et al., 2023; Yang et al., 2024; Huang et al., 2025; Xiang et al., 2025; Shu et al., 2025b;a). Despite these respective successes, both paradigms face fundamental bottlenecks that impede further progress:

1. **Numerical Models**: High-fidelity simulations require immense computational resources, often necessitating a trade-off between spatial resolution and computational efficiency (Cui et al., 2025; Xiong et al., 2023). This computational burden is particularly acute in regional downscaling simulations, where generating high-resolution boundary conditions from coarse global models is computationally prohibitive and sensitive to boundary inconsistencies. Furthermore, to maintain efficiency, models rely on simplified parameterizations for **missing physics**, introducing systematic biases that degrade long-term skill (Fox-Kemper et al., 2019).

2. **Data-Driven Models**: While pure AI models excel at capturing statistical patterns, they operate as "black boxes" blind to governing physical laws, frequently leading to violations of mass or momentum conservation and instability during long-term rollouts (Irrgang et al., 2021; Wu et al., 2025). Moreover, while deep learning has revolutionized spatial super-resolution in computer vision and meteorology, its application to dynamic ocean downscaling simulations—where phys-

ical consistency is paramount—remains a largely unexplored frontier.

This dichotomy prompts a fundamental question: *Can we forge a hybrid model where a lightweight physical model provides a stable dynamical skeleton, while a neural network acts as the adaptive flesh?* We address this by proposing a differentiable hybrid architecture. Unlike loose coupling, we embed deep neural networks directly into the time-stepping of a numerical solver, fusing rigorous physical laws with the expressivity of deep learning (Shu et al., 2025a; Gelbrecht et al., 2025; Kochkov et al., 2024).

We introduce **HybridOM**, a Global Hybrid Ocean Model that bridges rigorous fluid dynamics and modern machine learning. The main contributions are summarized as follows:

1. **Global Differentiable Hybrid Architecture**: We propose HybridOM, a novel framework that integrates a differentiable physical core with a multi-scale neural network. By synergizing the interpretability of physics with the fitting capability of AI, HybridOM achieves a high-precision, robust, and physics-consistent global ocean simulator, significantly outperforming baselines in long-term stability.

2. **SOTA Operational Forecasting System**: We construct a realistic global ocean forecasting system by coupling HybridOM with **FuXi-2.0** (Chen et al., 2023; Zhong et al., 2024), a state-of-the-art weather forecast model. Evaluated under standard operational protocols (El Aouni et al., 2026), our coupled system achieves state-of-the-art (SOTA) performance in 10-day global ocean forecasting.

3. **Effective Physics-Informed Downscaling**: We introduce a novel Flux Gating mechanism that utilizes the native thermodynamic fluxes from the dynamical core as a bridge between coarse global simulations and high-resolution regional dynamics. This approach effectively fills the gap in AI-driven regional ocean modeling.

## 2. Related Work

Current ocean forecasting methodologies are polarized between physics-based General Circulation Models (GCMs), which offer rigor but suffer from prohibitive computational costs (Kerbyson & Jones, 2005; Xu et al., 2015; Shchepetkin & McWilliams, 2005), and data-driven deep learning, which enhances efficiency but often violates physical laws (Cui et al., 2025; Wang et al., 2024; Xiong et al., 2023; Yang et al., 2024; Huang et al., 2025; Xiang et al., 2025; Shu et al., 2025b;a). While "gray-box" modeling—coupling

differentiable solvers with neural networks—has succeeded in atmospheric science and CFD (Kochkov et al., 2024; Gelbrecht et al., 2025; Frezat et al., 2022; Xu et al., 2024; Frezat et al., 2022; Bezgin et al., 2023; Xu et al., 2024), however, despite these advancements in related domains, these methods **cannot be directly applied to the ocean due to the distinct fluid properties and governing equations inherent to ocean dynamics**. What's more, **currently, there is no hybrid ocean model specifically designed for realistic, high-resolution ocean simulation or forecasting tasks.**

## 3. Method

In this section, we present the architecture of HybridOM by detailing its two foundational components. We first introduce the **Physical Skeleton**, a differentiable dynamical core, followed by the **Neural Flesh**, a multi-scale network designed to compensate for unresolved physics. We then describe the **Hybrid Integration Strategy**, which embeds the neural corrector directly into the time-stepping loop. Finally, we elaborate on our novel **Flux-Gated Downscaling** approach for regional simulations.

### 3.1. Global AI-Physics Hybrid Architecture

#### 3.1.1. PROBLEM DEFINITION

Global ocean modeling can be formalized as learning a state transition operator $\Phi$ that maps the current ocean state $\mathbf{X}_t$ (velocity, temperature, salinity, etc.) and atmospheric forcing $\mathbf{A}_{t:t+1}$ to the next state: $\mathbf{X}_{t+1} = \Phi(\mathbf{X}_t, \mathbf{A}_{t:t+1})$. Based on the source of $\mathbf{A}$, we distinguish between two critical tasks:

**Simulation (Hindcast):** Driven by observed history $\mathbf{A}_{t:t+1}^{obs}$ (e.g., ERA5), aiming to reconstruct physically consistent state $\mathbf{X}_{t+1}$.

**Forecasting (Coupled):** Driven by predicted forcing $\hat{\mathbf{A}}_{t:t+1}$ from a weather model (e.g., FuXi-2.0 (Zhong et al., 2024) in our study), aiming to predict future states $\mathbf{X}_{t+1}$ in an operational setting. It should be pointed out that FuXi-2.0 is not part of HybridOM's main architecture and is not jointly trained with HybridOM; it is used only in operational forecasting to provide future atmospheric forcing.

#### 3.1.2. HYBRIDOM: A LEARNABLE PDE SYSTEM

We conceptualize global ocean dynamics not merely as a discrete regression problem, but as a **learnable Evolutionary Partial Differential Equation (PDE)** system. We decompose the time evolution of the ocean state $\mathbf{X}$ a deterministic physical component based on known conservation laws (the "physical skeleton") and a data-driven, learnable residual component (the "neural flesh"). Formally, given the ocean state $\mathbf{X}$ and atmopheric forcing $\mathbf{A}$, the governing dynamics

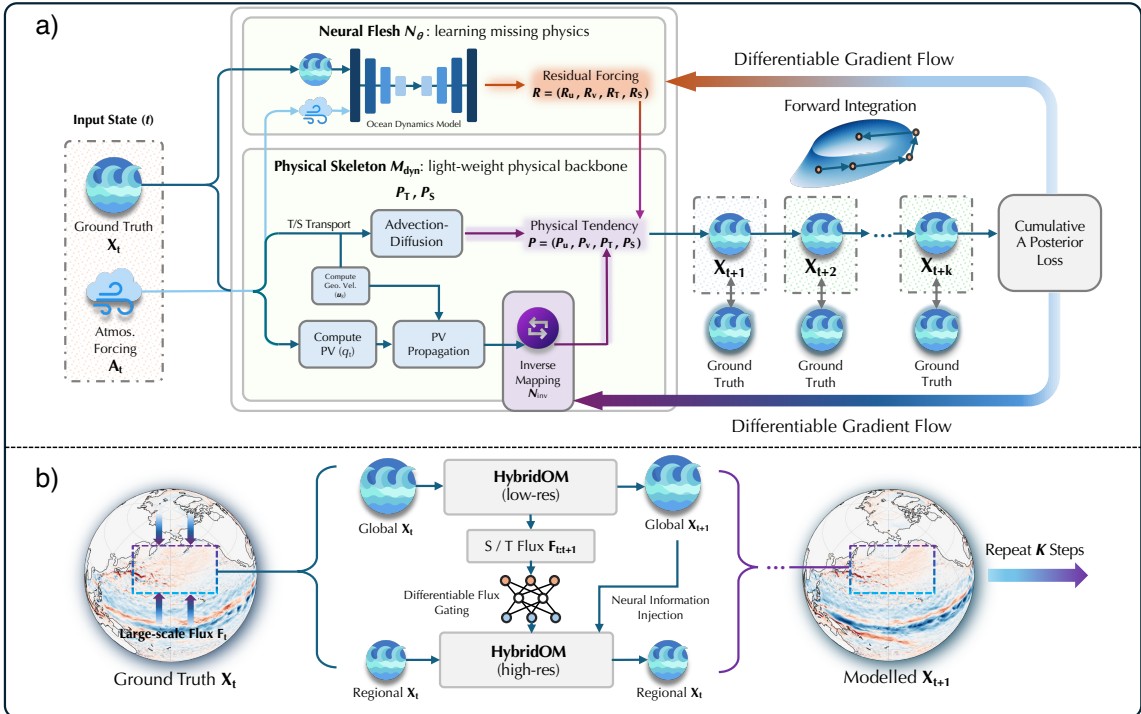

*Figure 1.* **Overview of the HybridOM Architecture.** (a) **Global Hybrid Framework.** The model couples a differentiable physical skeleton ($\mathcal{M}_{\text{phy}}$) for dynamical stability with a neural corrector ($\mathcal{N}_\theta$) to capture sub-grid physics. (b) **Spatial Downscaling.** This module employs a Flux Gating mechanism to inject coarse-scale thermodynamic constraints into high-resolution regional simulations.

are expressed as:

$$\frac{\partial \mathbf{X}}{\partial t} = \underbrace{\mathcal{M}_{\text{phy}}(\mathbf{X}, \mathbf{A})}_{\text{Physical Skeleton}} + \underbrace{\mathcal{N}_\theta(\mathbf{X}, \mathbf{A})}_{\text{Neural Flesh}}, \qquad (1)$$

where $\mathcal{M}_{\text{phy}}$ is the differentiable numerical solver enforcing conservation laws, and $\mathcal{N}_\theta$ captures subgrid turbulence and complex thermodynamic interactions through ODM (`OceanDynamicsModel`). The physical solver itself has no trainable neural parameters, except when it invokes the learnable inversion operator $\mathcal{N}_{\text{inv}}$ for velocity recovery; the residual corrector $\mathcal{N}_\theta$ is fully neural-parameterized.

To solve Eq. 1, we employ a differentiable time-stepping scheme. The discrete transition is obtained by integrating the hybrid tendencies:

$$\mathbf{X}_{t+1} = \text{Solver}\left(\mathbf{X}_t, \mathcal{M}_{\text{phy}}(\mathbf{X}_t, \mathbf{A}_{t:t+1}), \mathcal{N}_\theta(\mathbf{X}_t, \mathbf{A}_{t:t+1})\right). \tag{2}$$

Crucially, we adopt an *a posteriori* trajectory optimization strategy rather than one-step (Frezat et al., 2022; Kochkov et al., 2024). We unroll the solver for $K = 5$ days during training to minimize the cumulative forecast error:

$$\mathcal{L}(\theta) = \sum_{k=1}^{K} \left\| \hat{\mathbf{X}}_{t+k}(\theta) - \mathbf{X}_{t+k}^{\text{GT}} \right\|^2, \qquad (3)$$

Where $\hat{\mathbf{X}}_{t+k}$ is the forecasted ocean state while $\mathbf{X}_{t+k}^{\text{GT}}$ is the corresponding ground truth. By backpropagating gradients

through the differentiable solver ($\partial\hat{\mathbf{X}}/\partial\theta$), $\mathcal{N}_\theta$ learns to actively counteract the long-term numerical drift inherent in the coarse physical core. We set $K = 5$ to balance rollout accuracy and training cost; longer unrolls bring diminishing gains.

### 3.2. The Numerical Skeleton $\mathcal{M}_{\text{phy}}$: Differentiable Dynamics

The backbone of HybridOM is a differentiable solver operating on the 3-D ocean state variables $\mathbf{X} = \{u, v, T, S, \eta\}$, where $u, v$ are ocean current velocities, $T$ is temperature, $S$ is salinity and $\eta$ is sea surface height. To ensure stability and physical consistency, we decouple the dynamical evolution into three distinct processes: thermodynamic transport, potential vorticity propagation, and diagnostic surface adjustment.

**Thermodynamic Tracers** $(T, S)$. For potential temperature and salinity, the solver employs a strictly conservative flux-based formulation. Instead of the non-conservative advective form, we express the time evolution of any tracer $C \in \{T, S\}$ as the negative divergence of the total transport flux $\mathbf{F}_C$. The governing equation is given by:

$$\frac{\partial C}{\partial t} = -\nabla \cdot \mathbf{F}_C = -\nabla \cdot \left( \underbrace{\mathbf{u}C}_{\text{Advective Flux}} - \underbrace{\mathcal{K}_h \nabla C}_{\text{Diffusive Flux}} \right), \quad (4)$$

where $\mathbf{F}_C$ represents the total flux vector combining two distinct physical processes: the *advective flux* $\mathbf{u}C$, which transports properties along the resolved flow field, and the *diffusive flux* $-\mathcal{K}_h \nabla C$, which parameterizes irreversible mixing driven by background diffusivity $\mathcal{K}_h$. By formulating the dynamics in this flux-divergence form, the model ensures the conservation of heat and salt budgets globally (ignoring mixing and external forcing). Full discretization details are provided in Appendix B.

**Momentum Dynamics** $(u, v)$. Direct explicit integration of the primitive momentum equations is computationally prohibitive due to the stringent time-step constraints imposed by high-frequency gravity waves. To circumvent this, we adopt a simplified, implicit-style solver by projecting the dynamics into the Quasi-Geostrophic (QG) Potential Vorticity (PV) space (Vallis, 2017).

Specifically, we decouple the fast and slow manifolds via a "Diagnose-Evolve-Invert" cycle. First, we diagnose the geostrophic streamfunction $\psi$ and potential vorticity $q$ from the thermodynamic state $\{T, S, \eta\}$ (Appendix B). Then, in this reduced QG model, the flow evolution is governed by the conservation of potential vorticity, effectively filtering out noisy gravity waves:

$$\frac{\partial q}{\partial t} + \mathcal{J}(\psi, q) = \mathcal{F}_{\text{dissipation}}, \qquad (5)$$

where $\mathcal{J}(a, b) = \frac{\partial a}{\partial x}\frac{\partial b}{\partial y} - \frac{\partial a}{\partial y}\frac{\partial b}{\partial x}$ is the Jacobian operator. Finally, to close the loop, we map the evolved latent state back to the primitive velocity space through a learnable neural network $\mathcal{N}_{\text{inv}}$. This entire differentiable trajectory can be formalized as the following mapping flow:

$$\underbrace{\{S, T, \eta\}_t}_{\text{Input}} \xrightarrow{\text{Diagnose}} \underbrace{\{\psi, q\}_t}_{\text{QG space}} \xrightarrow[\text{Eq. 5}]{\text{Evolve}} \underbrace{\{q\}_{t+1}}_{\text{Next PV}}$$
$$\qquad \qquad \qquad \xrightarrow[\text{Decoder}]{\mathcal{N}_{\text{inv}}} \underbrace{\{u, v\}_{t+1}}_{\text{Output}} \qquad (6)$$

Here, $\mathcal{N}_{\text{inv}}$ serves as a **Learnable Inverse Laplace Operator**, approximating the inversion $\mathbf{u} \approx \mathbf{k} \times \nabla(\nabla^{-2}(q - \beta y))$ to recover the fine-scale velocity field (See Appendix B). The parameters of $\mathcal{N}_{\text{inv}}$ are optimized end-to-end together with ODM under the same multi-step trajectory loss.

**Sea Surface Height** $(\eta)$. In the physical skeleton, $\eta$ is not treated as a prognostic variable and is not evolved via the dynamical core. Instead, it directly participates in the diagnostic update of the velocity field $(u, v)$ through the pressure and geostrophic-balance terms (See Appendix B).

**Why the Skeleton is Lightweight.** The physical skeleton is lighter than full primitive-equation solvers because it combines conservative tracer transport with quasi-geostrophic PV evolution, avoiding explicit resolution of fast gravity-

wave modes. Unresolved turbulent and subgrid processes are then handled by the neural residual corrector $\mathcal{N}_\theta$.

### 3.3. The Neural Flesh $\mathcal{N}_\theta$: Ocean Dynamics Model

The `OceanDynamicsModel` (ODM) acts as a residual estimator $\mathcal{N}_\theta$, compensating for the simplified assumptions of the numerical skeleton. It maps the state $\mathbf{X}_t$ and forcing $\mathbf{A}_{t:t+1}$ to tendency corrections $\mathcal{R}_t$ via a hierarchical U-shaped architecture (Gao et al., 2022):

$$\begin{aligned} \mathbf{Z}_{\text{latent}} &= \text{Encoder}(\mathbf{X}_t \oplus \mathbf{A}_{t:t+1}), \\ \mathbf{Z}_{\text{evolved}} &= \text{LatentEvolution}(\mathbf{Z}_{\text{latent}}), \qquad (7) \\ \mathcal{R}_t &= \text{Decoder}(\mathbf{Z}_{\text{evolved}}) + \mathbf{Z}_{\text{skip}}. \end{aligned}$$

The core `LatentEvolution` module utilizes **Dual-Scale Ocean Attention (DSOA)** to efficiently resolve the dichotomy between local turbulence and global teleconnections. DSOA decomposes the feature space into two parallel branches:

**1. Local Branch (Windowed).** To capture fine-grained eddy interactions, we partition $\mathbf{Z}$ into non-overlapping windows of size $M \times M$ ($\mathcal{P}_{win}$). Self-attention is computed independently within each window, restricting the receptive field to local neighborhoods while maintaining high resolution. Cross-window communication is supplied by the global branch, encoder-decoder skip pathway, and final residual projection, which fuse local and global features before state correction.

**2. Global Branch (Grid).** To model basin-wide dependencies, we employ a grid partition strategy with stride $M$ ($\mathcal{P}_{grid}$). This operator gathers spatially sparse but globally distributed pixels into coarse-grained tokens, creating a dilated attention mechanism that covers the entire domain with linear complexity.

The multi-scale features from both branches are aggregated and fused through a residual projection layer:

$$\begin{aligned} \mathbf{Y}_{loc} &= \mathcal{P}_{win}^{-1}(\text{Attn}(\mathbf{Z}_{win})), \\ \mathbf{Y}_{glo} &= \mathcal{P}_{grid}^{-1}(\text{Attn}(\mathbf{Z}_{grid})), \qquad (8) \\ \mathbf{Z}_{out} &= \text{Proj}(\mathbf{Y}_{loc} \oplus \mathbf{Y}_{glo}) + \mathbf{Z}, \end{aligned}$$

where $\oplus$ denotes channel concatenation. This decoupled design enables HybridOM to simultaneously resolve sub-mesoscale turbulence and planetary-scale teleconnections within a single, computationally efficient block. Full architectural hyperparameters are provided in Appendix C.

### 3.4. Hybrid Integration and Time Stepping

We integrate the physical skeleton and neural flesh into a unified differentiable system. As discussed above, our framework instantiates two structurally identical but functionally distinct networks: the **Neural Flesh (Corrector)**

($\mathcal{N}_\theta$) and **Inversion Operator** ($\mathcal{N}_{\text{inv}}$). Details of the hyper-parameters can be found in Appendix E.

The forward integration from $t$ to $t + 1$ is formulated as:

$$\mathbf{X}_{t+1} = \mathcal{M}_{\text{phy}}\left(\mathbf{X}_t, \mathbf{A}_{t:t+1}; \mathcal{N}_{\text{inv}}\right) + \mathcal{N}_\theta(\mathbf{X}_t, \mathbf{A}_{t:t+1}). \quad (9)$$

The numerical solver $\mathcal{M}_{\text{phy}}$ first evolves the thermodynamic and PV fields and invokes $\mathcal{N}_{\text{inv}}$ to decode $(u, v)_{t+1}$. The residual corrector $\mathcal{N}_\theta$ then updates the physically evolved state. ERA5 forcing is used for simulation, while FuXi-2.0 forcing is used for operational forecasting. Gradients are backpropagated through the full computation graph to optimize $\mathcal{N}_{\text{inv}}$ and $\mathcal{N}_\theta$.

### 3.5. Regional Downscaling Simulation

We extend the HybridOM framework to regional down-scaling, formulating it as a boundary-value problem where the high-resolution state $\mathbf{X}_t^h$ evolves under the constraint of large-scale boundary conditions provided by a global parent model $\mathbf{X}_{t:t+1}^l$. To facilitate cross-scale interaction, following the approaches described in (Gao et al., 2025), we first extract a sub-domain from the global coarse output—slightly larger than the target simulation region to accommodate boundary inflows—and spatially interpolate it onto the high-resolution grid, denoted as $\hat{\mathbf{X}}_{t:t+1}^l$.

Our core innovation lies in leveraging the explicit flux formulations of the dynamical core to bridge scale interactions. We embed this coarse-to-fine fusion directly into the temporal integration via two coupled mechanisms: **Differentiable Flux Gating** and **Neural Information Injection**.

#### 3.5.1. DIFFERENTIABLE FLUX GATING (DFG)

To reconcile large-scale transport trends with local high-resolution dynamics, we operate explicitly in the physical flux space of variables $\{S, T\}$, as shown in Section 3.2. We propose a composite gating unit $\mathcal{N}_{\text{gate}}$ that fuses intrinsic fine-scale fluxes with interpolated coarse-scale fluxes through a two-stage mechanism: *Adaptive Soft-Gating* and *Residual Refinement*. Let $\mathcal{D}_{\text{flux}}$ denote the explicit flux operator. The process begins by computing the candidate fluxes from the fine physical skeleton ($\mathbf{F}^h$) and the upsampled coarse skeleton ($\hat{\mathbf{F}}^l$). These are then fused and refined to yield the final transport $\tilde{\mathbf{F}}_t$ and the subsequent state $\tilde{\mathbf{X}}_{t+1}$:

$$\mathbf{F}_t^h = \mathcal{D}_{\text{flux}}(\mathbf{X}_t^h), \quad \hat{\mathbf{F}}_{t:t+1}^l = \mathcal{D}_{\text{flux}}(\hat{\mathbf{X}}_{t:t+1}^l),$$
$$\mathbf{W} = \sigma\left(\mathcal{H}_{\text{sel}}\left(\mathbf{F}_t^h, \hat{\mathbf{F}}_t^l, \hat{\mathbf{F}}_{t+1}^l, \mathbf{A}\right)\right),$$
$$\mathbf{F}_{\text{pre}} = \mathbf{W}_h \odot \mathbf{F}_t^h + \mathbf{W}_{l,t} \odot \hat{\mathbf{F}}_t^l + \mathbf{W}_{l,t+1} \odot \hat{\mathbf{F}}_{t+1}^l, \quad (10)$$
$$\tilde{\mathbf{F}}_t = \mathbf{F}_{\text{pre}} + \mathcal{H}_{\text{ref}}\left(\mathbf{F}_{\text{pre}}, \mathbf{A}\right),$$
$$\tilde{\mathbf{X}}_{t+1} = \mathbf{X}_t^h + \Delta t \cdot \left(\nabla \cdot \tilde{\mathbf{F}}_t\right).$$

Here, $\sigma(\cdot)$ denotes Softmax along the channel dimension and $\odot$ denotes element-wise multiplication. The weights $\mathbf{W}$ are generated by the learnable selector $\mathcal{H}_{\text{sel}}$, rather than hand-specified, and adaptively fuse fine-grid, current coarse-grid, and next-step coarse-grid fluxes. The refiner $\mathcal{H}_{\text{ref}}$ then adds a residual flux correction before the divergence operator is applied. Full implementation details are provided in Appendix C.

#### 3.5.2. NEURAL INFORMATION INJECTION (NII)

While flux gating ensures thermodynamic stability, it does not fully account for the sub-grid kinematic effects driven by large-scale variations. To address this, we explicitly inject the large-scale context into the "Neural Flesh" module. As described in Section 3.3, the `OceanDynamicsModel` ($\mathcal{N}_{\text{dyn}}$) serves as a residual corrector, taking both the physically evolved intermediate state $\tilde{\mathbf{X}}_{t+1}^h$ and the interpolated global context $\hat{\mathbf{X}}_{t+1}^l$ as inputs:

$$\mathbf{X}_{t+1}^h = \tilde{\mathbf{X}}_{t+1}^h + \mathcal{N}_\theta\left(\tilde{\mathbf{X}}_{t+1}^h \oplus \hat{\mathbf{X}}_{t+1}^l, \mathbf{A}_t\right), \quad (11)$$

where $\oplus$ denotes channel concatenation and $\tilde{\mathbf{X}}_{t+1}^h$ is the intermediate regional state evolved by the flux-gated physical update.

The downscaling solver is trained end-to-end by unrolling the integrator for $K = 5$ steps and minimizing a MSE loss against high-resolution analysis data $\mathbf{X}_{\text{GT}}^h$.

## 4. Experiments

We evaluate HybridOM on the GLORYS12V1 reanalysis (Jean-Michel et al., 2021) and OceanBench (El Aouni et al., 2026) dataset, focusing on its ability to model complex ocean dynamics across varying spatial scales and temporal horizons. Our evaluation is structured to assess both the long-term subseasonal-to-seasonal stability of the model and its short-term operational forecasting skill under realistic atmospheric coupling.

### 4.1. Experimental Setup

To rigorously evaluate the capabilities of HybridOM, we design a comprehensive evaluation framework consisting of three distinct experimental regimes: *Global Simulation*, *Global Forecasting*, and *Regional Downscaling*.

**Dataset.** Our dataset is constructed from CMEMS products (Jean-Michel et al., 2021; El Aouni et al., 2026) and ERA5 atmospheric forcing (Hersbach et al., 2020), tailored to diverse experimental scales. For historical simulations (1993–2020), all models are trained on a unified $0.5°$ global grid derived from the GLORYS12V1 reanalysis. In forecasting tasks, we evaluate both $0.5°$ and $0.25°$ variants of HybridOM, using GLO12 Nowcast for initialization and

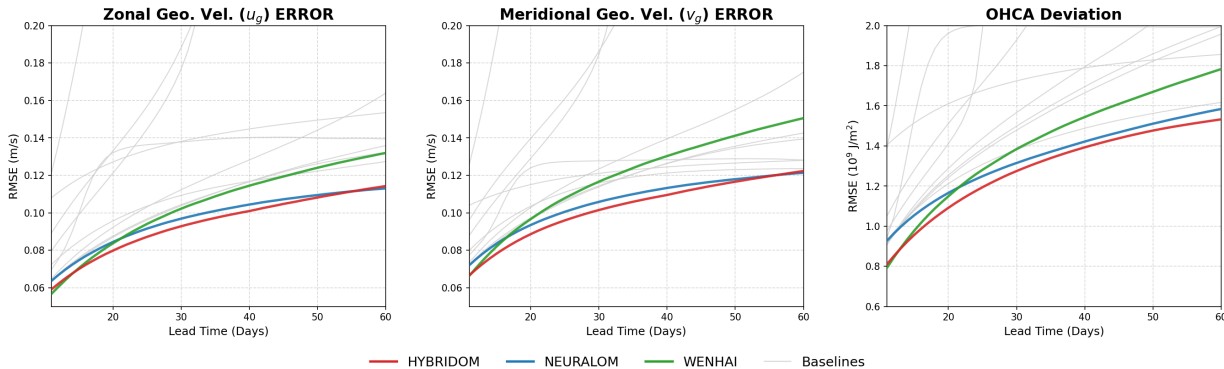

*Figure 2.* **Global Simulation Error Analysis.** Comparison of HybridOM against baselines. From left to right, the panels illustrate the errors in **Zonal** and **Meridional Geostrophic Velocities**, followed by the deviation in **Upper Ocean Heat Content (OHC)**.

*Table 1.* **Global Ocean Simulation Performance.** Comparison of HybridOM against top-performing baselines on GLORYS dataset. We report Weighted MSE (lower is better) for $T, S, U, V$ at **15-day** and **40-day** horizons. Additionally, we report the 30-day CSI (higher is better) for SSTA (Sea Surface Temperature Anomaly). **Bold**: Best; Underlined: Second Best. Full baseline results can be found in Appendix G.1.2.

| Model | wMSE $T$ | | wMSE $S$ | | wMSE $U$ | | wMSE $V$ | | CSI Score (SSTA) | | |
|---|---|---|---|---|---|---|---|---|---|---|---|
| | 15-day | 40-day | 15-day | 40-day | 15-day | 40-day | 15-day | 40-day | 90% | 92.5% | 95% |
| 📷 U-Net | 0.3576 | 0.6546 | 0.0235 | 0.0390 | 0.0104 | 0.0182 | 0.0101 | 0.0156 | 0.5412 | 0.5285 | 0.5124 |
| 📷 DiT | 0.3258 | 0.8106 | 0.0208 | 0.0597 | 0.0084 | 0.0209 | 0.0089 | 0.0203 | 0.6268 | 0.6173 | 0.6050 |
| ☁ GraphCast | 0.3321 | 0.8817 | 0.0224 | 0.0857 | 0.0085 | 0.0224 | 0.0088 | 0.0207 | 0.5774 | 0.5684 | 0.5570 |
| ☁ OneForecast | 0.3318 | 0.9134 | 0.0215 | 0.0992 | 0.0081 | 0.0208 | 0.0085 | 0.0193 | 0.5964 | 0.5870 | 0.5753 |
| 🌊 WenHai | 0.2805 | 0.7236 | 0.0202 | 0.0486 | 0.0093 | 0.0221 | 0.0087 | 0.0182 | 0.6419 | 0.6309 | 0.6167 |
| 🌊 NeuralOM | 0.2768 | 0.5284 | 0.0168 | 0.0307 | 0.0077 | 0.0139 | 0.0078 | 0.0135 | 0.6393 | 0.6311 | 0.6204 |
| 🏆 HybridOM | **0.2392** | **0.4958** | **0.0155** | **0.0291** | **0.0064** | **0.0121** | **0.0068** | **0.0126** | **0.6701** | **0.6620** | **0.6520** |

GLO12 Analysis for validation throughout 2024. For regional experiments, a cropped $0.25°$ subdomain from the Northwest Pacific is utilized as the high-resolution training target. Details of the data source, pre and postprocessing can be found in Appendix D.

The ocean state $\mathbf{X} \in \mathbb{R}^{93 \times H \times W}$ comprises 2D Sea Surface Height $\eta$ and four 3D variables $(u, v, T, S)$ across the top 23 vertical layers (surface to ~644 m). This is driven by an atmospheric forcing tensor $\mathbf{A} \in \mathbb{R}^{3 \times H \times W}$, representing 10m zonal wind $(u_{10})$, 10m meridional wind $(v_{10})$, and 2m air temperature $(t_{2m})$. These drivers are sourced from ERA5 reanalysis for simulation and 10-day rollouts from the FuXi-2.0 weather model for forecasting. Detailed description are provided in Appendix D.

**Task Configurations.** First, in the **Global Simulation (Hindcast)** task, we utilize ground truth atmospheric forcing from ERA5 to drive the model for long-term, full-variable global simulation. In this regime, we extensively benchmark HybridOM against a wide range of state-of-the-art models spanning oceanography, meteorology, computer vision (CV), and spatio-temporal (ST) forecasting domains (Gao

et al., 2026; Cui et al., 2025; Kurth et al., 2023; Liu et al., 2025; Ronneberger et al., 2015; He et al., 2016; Peebles & Xie, 2023; Shi et al., 2015; Gao et al., 2022; Wu et al., 2024). All experiments in this task is conducted under $0.5°$ resolution. Second, the **Global Forecasting** task assesses operational viability by coupling HybridOM with the FuXi-2.0 weather model. Driven by the *predicted* atmospheric forcing from FuXi-2.0, we evaluate the system strictly following the standardized protocols of **OceanBench** (El Aouni et al., 2026). Third, the **Regional Downscaling** task focuses on regional high resolution simulation forcing by global low resolution information.

**Metrics.** We comprehensively evaluate model performance across three critical dimensions: **General Accuracy:** We employ standard Latitude-Weighted Root Mean Square Error (wRMSE) and Mean Square Error (wMSE) to quantify the overall precision of the simulation against ground truth. **Extreme Event Capture:** To assess performance in extreme scenarios (e.g., Marine Heatwaves (Holbrook et al., 2019; Oliver et al., 2021)), we utilize the Critical Success Index (**CSI**). A higher CSI indicates a superior capability to resolve sharp, realistic extremes rather than yielding

conservative, blurred predictions. **Physical Consistency:** Following OceanBench protocols (El Aouni et al., 2026), we validate dynamical fidelity using Geostrophic Current Mean Error, Lagrangian Trajectory Error (LTE) and Upper Ocean Heat Content Deviation (OHC). (See Appendix E for definitions).

**Implementation Details.** All models are implemented in PyTorch and trained on a cluster of 8 NVIDIA A100 GPUs using Distributed Data Parallel (DDP). Comprehensive hyperparameter settings are provided in Appendix E.

## 4.2. Global Ocean Simulation Performance

We evaluate the long-term simulation accuracy and physical realism of HybridOM by conducting a 60-day integration during 2020 under $0.5°$ resolution. Table 1 and Figure 2 illustrates the key performance metrics compared against the GLORYS12 reanalysis. Experimental results demonstrate that within the 10-to-60-day window, HybridOM achieves state-of-the-art (SOTA) general accuracy across all prognostic variables in terms of weighted MSE. Beyond statistical metrics, our model also exhibits SOTA performance in capturing extreme events, specifically in simulating the evolution of Marine Heatwaves (i.e., the CSI result of SSTA in Table 1). Furthermore, HybridOM shows the highest physical consistency in resolving geostrophic balance and upper-ocean heat content. We attribute this robustness to two key architectural designs: first, the momentum component of our dynamical core is grounded in Quasi-Geostrophic (QG) theory, which intrinsically preserves geostrophic balance information; second, the governing equations of scalar tracers (temperature and salinity) are formulated in flux-form, enabling a more precise representation of the sources and sinks within the upper-ocean heat budget.

We further verify the simulation against sparse in-situ observations. Following the observation-based probabilistic evaluation protocol of WenHai (Cui et al., 2025), Figure 3 summarizes the 60-day pesudo Continuous Ranked Probability Score (CRPS) on drifter Sea Surface Temperature (SST) and Argo temperature/salinity profiles, where HybridOM achieves the lowest CRPS across all three observation-based metrics. As an additional physical sanity check, Appendix Figure 9 reports surface kinetic/enstrophy spectra; HybridOM preserves spectral fidelity comparable to strong baselines.

## 4.3. Ocean Forecasting Performance

In operational forecasting tasks, maintaining accuracy under atmospheric forcing uncertainty is important. Following OceanBench protocols, we evaluate HybridOM by coupling it with the FuXi-2.0 weather model and benchmarking against state-of-the-art baselines (see Appendix A for details), including GLONET (El Aouni et al., 2025), WenHai

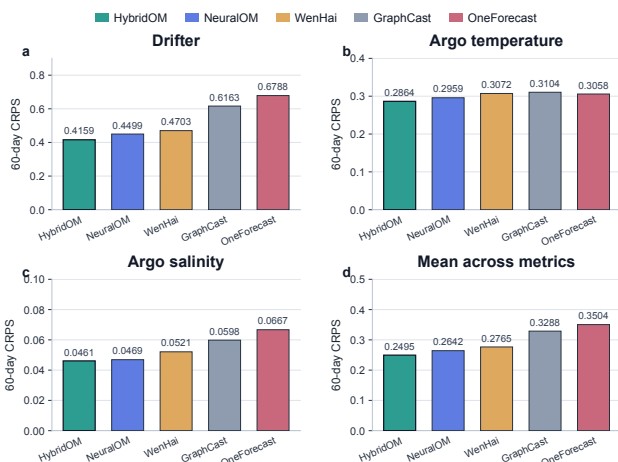

*Figure 3.* **Observation-based pseudo CRPS at 60 days.** (a) Drifter CRPS of Sea Surface Temperature (SST), (b) Argo temperature-profile CRPS, (c) Argo salinity-profile CRPS, and (d) the mean CRPS across the three observation-based metrics. Lower values indicate better probabilistic consistency with independent in-situ observations.

(Cui et al., 2025), and XiHe (Wang et al., 2024). To ensure a fair comparison, the forecast outputs from all models—including the baselines—are spatially interpolated to a unified $1/12°$ resolution and validated against the GLO12 Analysis. As summarized in Figure 4, HybridOM achieves SOTA performance across most surface and 220m metrics, while also yielding competitive or best LTE across the evaluated regions. Notably, HybridOM-0.25 excels in short-range forecasting ($\leq 3$ days), while HybridOM-0.5 is more effective at longer horizons.

## 4.4. Spatial Downscaling Results

To validate our flux-reconstruction-based downscaling, we conduct comparative experiments in the high-resolution North Pacific domain. As specialized baselines for regional ocean simulation are scarce, we compare HybridOM against the **Neural Nested Grid (NNG)** proposed in OneForecast (Gao et al., 2025) and **GraphEFM** (Oskarsson et al., 2024). Furthermore, we evaluate the contribution of our specific architectural components—Differentiable Flux Gating (DFG) and Neural Information Injection (NII)—through an extensive ablation study (Figure 5). A detailed discussion of these performance gains and module contributions is provided in Section 4.5.

Interestingly, while the Flux Gating mechanism explicitly targets thermodynamic fluxes ($T, S$), Figure 5 reveals concurrent improvements in dynamic variables ($U, V$). We attribute this cross-variable benefit to the accurate modeling of thermodynamic-dynamic feedback through $\mathcal{N}_\theta$.

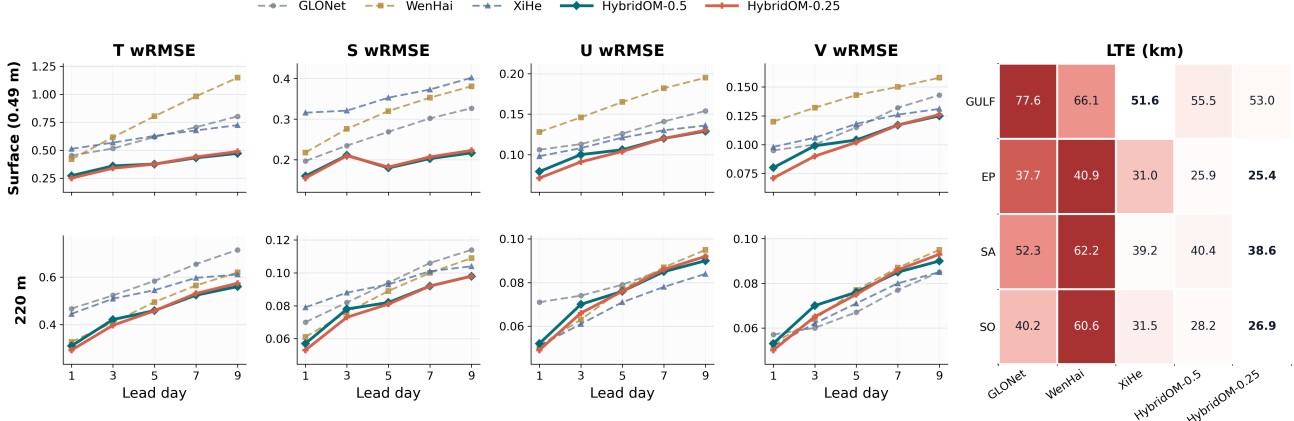

*Figure 4.* **Global Ocean Forecasting wRMSE and Lagrangian Trajectory Error (LTE).** Comparison of HybridOM variants ($0.5°$ and $0.25°$) against SOTA baselines. The left panels report weighted RMSE for surface and 220m depths across lead times of 1, 3, 5, 7, and 9 days; the right panel reports 9-day Lagrangian trajectory errors across four regions (GULF, EP, SA, SO). Lower values indicate better performance.

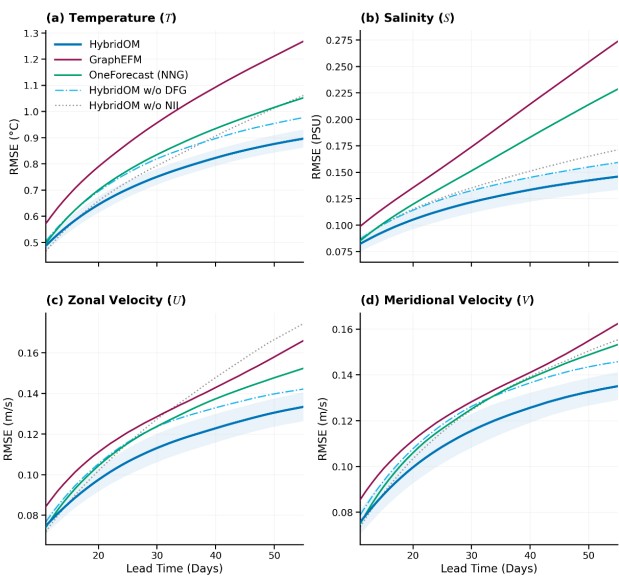

*Figure 5.* **Stability and Ablation Analysis of Spatial Downscaling.** Time evolution of Latitude-Weighted RMSE for regional simulation in the North Pacific based on different models.

## 4.5. Ablation Studies

We analyze HybridOM by decoupling its core components: the global physical-neural hybrid and the regional downscaling constraints. We evaluate the model performance by reporting the **thermodynamic error** (defined as the sum of the weighted MSE for salinity $S$ and temperature $T$) and the **dynamic error** (defined as the sum of the weighted MSE for zonal velocity $u$ and meridional velocity $v$).

**Impact of the Physical Skeleton.** We first isolate the role of the numerical solver by comparing HybridOM against a pure neural baseline, including S/T solver and U/V solver (Section 3.2). As shown in Table 2, the physical skeleton yields minimal gains in short-term forecasts (Day 10). However, its importance grows with time; removing physics leads to long-term drift by Day 50, especially for velocity fields when removing U/V solver. This confirms that while the neural component handles immediate precision, the physical core is essential for long-term regularization.

**Neural Architecture Design.** Next, we isolate the core design of our neural flesh: Local Branch, Global Branch and a shallower `LatentEvolution`. Ablation experiments reveal that the design of the neural component primarily dictates short-term predictive skill (e.g., at the 10-day horizon). Conversely, for long-term simulations (e.g., 50 days), the impact of neural architecture variations diminishes as the physical skeleton becomes the dominant factor in preventing trajectory drift. Furthermore, results indicate that the Local and Global attention branches are of equal importance in capturing multi-scale dynamics. Interestingly, the framework exhibits remarkable robustness to model depth; even when significantly reducing the `LatentEvolution` from 8 to 2 layers, HybridOM maintains competitive performance.

**Impacts on Physical Consistency.** Beyond wMSE, we also evaluate the influence of different architectural components on physical consistency metrics (see the lower part of Table 2). Compared to the neural network components, the physical core has a larger impact on these physical metrics. This observation aligns with our premise, highlighting the role of physical priors in maintaining the structural integrity of ocean dynamics.

**Regional Constraints.** Finally, for regional downscaling, as shown in Figure 5, we find that both Differentiable Flux Gat-

*Table 2.* **Ablation Summary.** The first block reports wMSE for thermodynamic variables $(S + T)$ and velocity variables $(U + V)$; the second block reports 50-day OHCA $(10^9 \mathrm{J/m^2})$ and regional LTE (km) diagnostics, where LTE-P and LTE-A denote Lagrangian Trajectory Error in the Pacific and Atlantic basins, respectively. Lower values are better.

| | Global Simulation wMSE | | | |
| --- | --- | --- | --- | --- |
| **Config.** | **10d** | | **50d** | |
| | $S+T$ | $U+V$ | $S+T$ | $U+V$ |
| **HybridOM** | **0.1793** | **0.0095** | **0.5946** | **0.0278** |
| *Physical Core* | | | | |
| w/o S/T | 0.1803 | 0.0095 | 0.6240 | 0.0287 |
| w/o U/V | 0.1808 | 0.0095 | 0.5998 | 0.0321 |
| *Neural Flesh* | | | | |
| w/o Glo. | 0.1880 | 0.0099 | 0.6146 | 0.0281 |
| w/o Loc. | 0.1871 | 0.0100 | 0.6101 | 0.0290 |
| Shallow | 0.1895 | 0.0103 | 0.6134 | 0.0292 |

| | Physical Metrics, 50-day | | |
| --- | --- | --- | --- |
| **Variant** | **LTE-P** | **LTE-A** | **OHCA** |
| **HybridOM** | **147.1** | **200.5** | **1.477** |
| *Physical Core* | | | |
| w/o S/T | 178.9 | 217.5 | 1.610 |
| w/o U/V | 176.4 | 233.1 | 1.656 |
| *Neural Flesh* | | | |
| w/o Glo. | 164.9 | 211.9 | 1.518 |
| w/o Loc. | 166.9 | 218.9 | 1.515 |
| Shallow | 162.2 | 212.1 | 1.520 |

ing (DFG) and Neural Information Injection (NII) contribute positively to the long-term simulation stability. Specifically, the DFG mechanism demonstrates more pronounced effectiveness during the medium-range window (Day 10–30). In contrast, for longer horizons (beyond Day 30), the NII module—which directly injects large-scale context via $\mathcal{N}_\theta$—yields stronger performance gains.

### 4.6. Additional Notes

We provide some additional results and discussions on efficiency, scaling and sensitivity.

**Notes on Efficiency and Scaling.** The efficiency and scaling analyses in Appendices G.1.1 and G.1.5 place this accuracy-cost trade-off in context. Although HybridOM is not the lowest-latency neural surrogate and requires more computation than compact neural baselines, it substantially reduces the computational burden relative to strong ocean and atmospheric predictors, including NeuralOM (Gao et al., 2026), GraphCast (Lam et al., 2023), and OneForecast (Gao et al., 2025). For comparison with traditional numerical modeling, POM (the Princeton Ocean Model) is a classical physics-based ocean circulation model that solves the three-dimensional primitive equations for ocean thermodynamics

and dynamics (Xu et al., 2015). Under the same configuration, HybridOM runs $211.20\times$ and $271.21\times$ faster than GPU-accelerated explicit and implicit POM solvers, respectively, with a one-day inference time of 510.76 ms (Table 13 in Appendix). This advantage reflects the role of the lightweight physical skeleton, which retains large-scale balance while leaving unresolved subgrid effects to the learned correction. When it comes to high-resolution scaling, the bottleneck mainly comes from the fixed-width neural components rather than the dynamical core. Directly preserving the original neural configuration at $1/12°$ is therefore prohibitively expensive. Increasing the spatial downsampling depth of both $\mathcal{N}_\theta$ and $\mathcal{N}_{\mathrm{inv}}$ provides a practical solution: with 10 downsampling layers, representative inference memory, training memory, and inference time are reduced by $12.6\times$, $17.9\times$, and $5.9\times$, respectively, relative to the naive $1/12°$ full model (see Appendix G.1.5).

**Initial-Condition Sensitivity.** A useful ocean simulator should respond to changes in the ocean state without turning small initialization differences into scale-dependent numerical artifacts. We therefore measure a normalized finite-difference response: the initial ocean state is perturbed, the atmospheric forcing is kept fixed, and the rollout difference is normalized by the size of the initial perturbation (Appendix G.1.3). The response grows moderately from Day 1 to Day 10, but remains nearly identical when the perturbation amplitude increases from $\varepsilon_{\mathrm{IC}} = 0.01$ to $\varepsilon_{\mathrm{IC}} = 0.05$. This indicates that HybridOM retains meaningful dependence on ocean initial conditions while avoiding artificial amplification tied to the perturbation scale.

## 5. Conclusion

This work introduces HybridOM, a novel framework bridging numerical oceanography and deep learning via a differentiable physical "skeleton" and a neural "flesh." By integrating strict physical laws with data-driven flexibility, HybridOM effectively mitigates the long-standing trade-off between computational efficiency and physical fidelity. Across tasks ranging from global simulation and operational forecasting to regional downscaling, our model achieves state-of-the-art accuracy while preserving rigorous physical consistency and long-term stability. These results establish HybridOM as a scalable and interpretable foundation for next-generation ocean digital twins.

## Acknowledgements

This work was supported by the National Natural Science Foundation of China (42125503, 42430602).

## Impact Statement

This paper presents work whose goal is to advance the field of Machine Learning. There are many potential societal consequences of our work, none which we feel must be specifically highlighted here.

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

# A. Algorithm

## A.1. Overall Framework

We outline the training and inference logic of the Global Simulation, Global Forecasting, and Regional Downscaling procedures. The detailed algorithms of global and regional simulation are listed in Algorithm 1.

What's more, in the **Global Forecasting** task, we adopt a coupled approach with FuXi-2.0: the weather model first generates predicted atmospheric states for the subsequent 24-hour window (at UTC 12:00 for $t + 1$). This predicted field is then concatenated with the forcing field of the current time $t$ to serve as the external drive for HybridOM's forward integration. The detailed operational workflow for this coupled forecasting is illustrated in Figure 6.

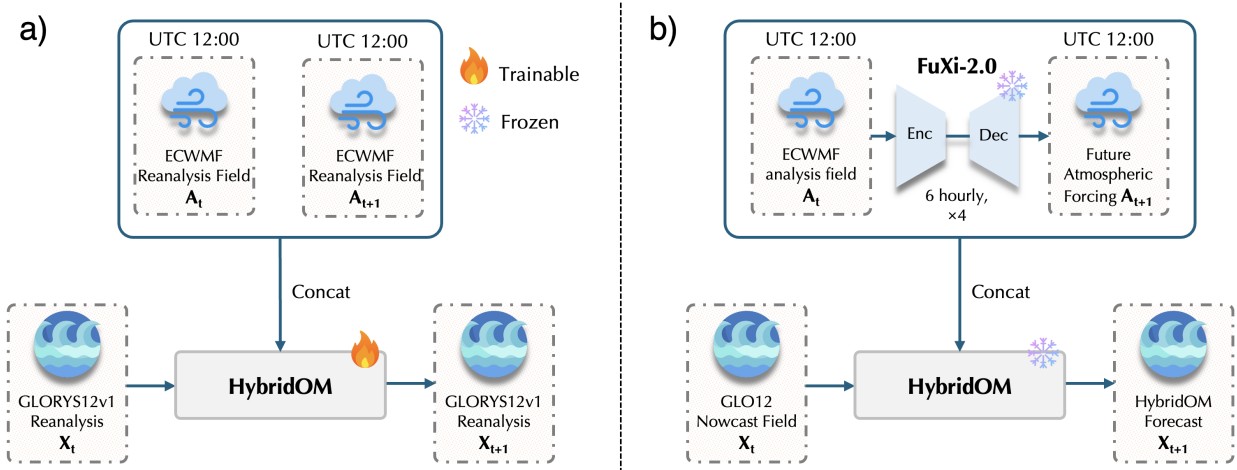

*Figure 6.* **HybridOM Operational Workflow.** The diagram illustrates the coupling between FuXi-2.0 and HybridOM for global forecasting.

# B. Details of the Dynamical Core

The physical skeleton of HybridOM is a differentiable Finite Volume solver implemented in PyTorch. It explicitly solves the primitive equations for tracers and vorticity on a spherical grid, while diagnosing velocity and pressure to maintain geostrophic balance.

## B.1. Governing Equations

The system evolves the state vector $\mathbf{X} = \{T, S, \zeta\}$ by coupling thermodynamic transport with quasi-geostrophic dynamics. We categorize the governing laws into three coupled groups: **Thermodynamics** (blue), **Vorticity Dynamics** (red), and **Diagnostic Closure** (green).

$$\text{Tracer Transport:} \quad \frac{\partial C}{\partial t} + \nabla \cdot (\mathbf{u}C) = \nabla \cdot (\mathcal{K}_h \nabla C), \quad C \in \{T, S\} \tag{12}$$

$$\text{Vorticity Evolution:} \quad \frac{\partial \zeta}{\partial t} + \nabla \cdot (\mathbf{u}_g \zeta) = \nabla \cdot (A_h \nabla \zeta) - \beta v \tag{13}$$

$$\text{Diagnostic Closure:} \quad \begin{cases} \rho(z) = \text{EOS}(T, S, z) \\ p(z) = g\rho_0 \eta + \int_z^0 g(\rho(z') - \rho_0)dz' \\ \mathbf{u}_g = \hat{\mathbf{k}} \times \nabla p/(\rho_0 f) \\ \zeta = \nabla \times \mathbf{u} \approx \frac{\nabla^2 p}{\rho_0 f} \end{cases} \tag{14}$$

---

**Algorithm 1** HybridOM: Training Procedures for Global and Regional Tasks

---

1: **Procedure 1: Global Ocean Simulation (Training)**

**Input:** Initial state $\mathbf{X}_0$, Forcing $\mathbf{A}_{0:K}$, Ground Truth $\mathbf{X}_{1:K}^{GT}$

**Input:** Physical Skeleton $\mathcal{M}_{\text{phy}}$, Neural Flesh $\mathcal{N}_\theta$, Inversion Net $\mathcal{N}_{\text{inv}}$

2: $\quad \hat{\mathbf{X}}_0 \leftarrow \mathbf{X}_0, \mathcal{L}_{gl} \leftarrow 0$

3: **for** step $t = 0$ to $K - 1$ **do**

4: $\quad$ // 1. Physical Skeleton Evolution (AB-II Scheme)

5: $\quad \tilde{\mathbf{X}}_{t+1} \leftarrow \mathcal{M}_{\text{phy}}(\hat{\mathbf{X}}_t, \mathbf{A}_t; \mathcal{N}_{\text{inv}})$

6: $\quad$ // 2. Neural Residual Injection

7: $\quad \mathcal{R}_t \leftarrow \mathcal{N}_\theta(\hat{\mathbf{X}}_t, \mathbf{A}_t)$ {Corrects $\hat{\mathbf{X}}_t$ based tendencies}

8: $\quad \hat{\mathbf{X}}_{t+1} \leftarrow \tilde{\mathbf{X}}_{t+1} + \mathcal{R}_t$

9: $\quad$ // 3. Global Loss Accumulation

10: $\quad \mathcal{L}_{gl} \leftarrow \mathcal{L}_{gl} + \|\hat{\mathbf{X}}_{t+1} - \mathbf{X}_{t+1}^{GT}\|^2$

11: **end for**

12: Update $\{\theta, \mathcal{N}_{\text{inv}}\}$ via $\nabla \mathcal{L}_{gl}$

13: **End Procedure**

---

14: **Procedure 2: Regional Downscaling (Training)**

**Input:** Regional IC $\mathbf{x}_0^h$, Coarse Context $\mathbf{X}_{0:T}^l$, Regional GT $\mathbf{x}_{1:T}^{h,GT}$

**Input:** Gate $\mathcal{N}_{\text{gate}}$, Regional Neural Flesh $\mathcal{N}_\theta$

15: $\hat{\mathbf{x}}_0^h \leftarrow \mathbf{x}_0^h, \mathcal{L}_{reg} \leftarrow 0$

16: **for** step $t = 0$ to $T - 1$ **do**

17: $\quad$ // 1. Differentiable Flux Gating (DFG)

18: $\quad \mathbf{f}_t^h \leftarrow \text{GetFlux}(\hat{\mathbf{x}}_t^h), \quad \hat{\mathbf{F}}_t^l \leftarrow \text{GetFlux}(\text{Interp}(\mathbf{X}_t^l))$

19: $\quad \tilde{\mathbf{f}}_t \leftarrow \mathcal{N}_{\text{gate}}(\mathbf{f}_t^h, \hat{\mathbf{F}}_t^l)$ {Eq. 10}

20: $\quad$ // 2. Constrained Temporal Evolution

21: $\quad \tilde{\mathbf{x}}_{t+1}^h \leftarrow \text{Evolve}(\hat{\mathbf{x}}_t^h, \tilde{\mathbf{f}}_t)$ {Constrained by fused fluxes}

22: $\quad$ // 3. Neural Information Injection (NII)

23: $\quad \mathbf{C}_{t+1} \leftarrow \tilde{\mathbf{x}}_{t+1}^h \oplus \text{Interp}(\mathbf{X}_{t+1}^l)$

24: $\quad \hat{\mathbf{x}}_{t+1}^h \leftarrow \tilde{\mathbf{x}}_{t+1}^h + \mathcal{N}_\theta(\mathbf{C}_{t+1}, \mathbf{A}_t)$ {Eq. 11}

25: $\quad$ // 4. Regional Loss Accumulation

26: $\quad \mathcal{L}_{reg} \leftarrow \mathcal{L}_{reg} + \|\hat{\mathbf{x}}_{t+1}^h - \mathbf{x}_{t+1}^{h,GT}\|^2$

27: **end for**

28: Update $\{\theta, \mathcal{N}_{\text{gate}}\}$ via $\nabla \mathcal{L}_{reg}$

29: **End Procedure**

---

Equations 12–14 describe the governing dynamics. The variables are defined as follows:

- **Prognostic Variables:** $C$ denotes tracers (Temperature $T$ and Salinity $S$); $\zeta$ is the relative vorticity, defined as the curl of velocity ($\nabla \times \mathbf{u}$) and approximated by the Laplacian of pressure ($\nabla^2 p$) scaled by $\rho_0 f$ under geostrophic balance.

- **Diagnostic Variables:** $\rho$ is density from the Equation of State (EOS); $p$ is hydrostatic pressure (sum of barotropic and baroclinic terms); $\mathbf{u}_g$ is the geostrophic velocity.

- **Forcing & Parameters:** $\beta v$ represents the planetary vorticity advection (beta-effect), where $v$ is meridional velocity; $f$ is the Coriolis parameter; $\mathcal{K}_h$ and $A_h$ are horizontal diffusivity and viscosity coefficients; $\hat{\mathbf{k}}$ is the vertical unit vector.

Eq. 13 evolves the relative vorticity $\zeta = \nabla \times \mathbf{u}$, avoiding the fast gravity waves inherent in primitive momentum equations. By introducing the neural flesh to correct the missing physics, this system is closed at each time step.

**B.2. Numerical Implementation and Dynamical Core**

**Spatial Discretization and Flux Reconstruction.** In this study, we employ a staggered Arakawa C-grid topology (Arakawa & Lamb, 1977). Scalar fields—potential temperature ($T$), salinity ($S$), density ($\rho$), and sea surface height ($\eta$)—are defined

at cell centers $(i, j)$, while zonal $(u)$ and meridional $(v)$ velocities are staggered at the east-west $(i \pm 1/2, j)$ and north-south $(i, j \pm 1/2)$ cell faces, respectively. This arrangement naturally suppresses spurious pressure-velocity decoupling (checkerboard modes) and facilitates the accurate computation of the discrete divergence operator.

The evolution of scalar tracers $\phi \in \{\zeta, T, S\}$ is governed by the conservation law $\partial_t \phi = -\nabla \cdot (\mathbf{F}^{\text{adv}} + \mathbf{F}^{\text{diff}})$, where the total flux consists of an advective component and a diffusive component. We discretize these terms on the staggered Arakawa C-grid to ensure rigorous mass conservation.

*1) Advective Flux.* We implement a differentiable **First-Order Upwind Scheme** for advective flux reconstruction (Courant et al., 1952). While lower-order schemes are numerically diffusive, they guarantee monotonicity and numerical stability, which is critical for maintaining stable gradients during end-to-end backpropagation. The advective flux $F^{\text{adv}}$ at the cell interface $(i + 1/2)$ is determined by the upstream scalar value:

$$F_{i+1/2}^{\text{adv}} = \begin{cases} u_{i+1/2} \cdot \phi_i & \text{if } u_{i+1/2} > 0, \\ u_{i+1/2} \cdot \phi_{i+1} & \text{otherwise.} \end{cases} \tag{15}$$

In our implementation, this conditional logic is executed via differentiable masking operations (i.e., `torch.where`). Crucially, unlike non-differentiable index lookups, these operators preserve the computational graph, ensuring that gradients propagate exactly through the active physical branch back to both the velocity field $u$ and the tracer distribution $\phi$ without requiring smooth approximations.

*2) Diffusive Flux.* To parameterize sub-grid mixing and suppress grid-scale noise, we model the diffusive flux using Fickian diffusion discretized via a **Second-Order Central Difference** scheme:

$$F_{i+1/2}^{\text{diff}} = -\mathcal{K}_h \frac{\phi_{i+1} - \phi_i}{\Delta x}, \tag{16}$$

where $\mathcal{K}_h$ is the horizontal Laplacian eddy diffusivity. The final tendency for the cell $(i)$ is computed as the divergence of the net flux:

$$\frac{\partial \phi_i}{\partial t} = -\frac{(F^{\text{adv}} + F^{\text{diff}})_{i+1/2} - (F^{\text{adv}} + F^{\text{diff}})_{i-1/2}}{\Delta x}. \tag{17}$$

**Temporal Integration Strategy.** The prognostic equations are stepped forward using a **Quasi-Second-Order Adams-Bashforth (AB2)** scheme. To mitigate the computational mode inherent to multi-step methods, we introduce a frequency-dependent damping term controlled by the off-centering parameter $\epsilon$. The update rule for any state variable $\psi \in \{\zeta, T, S\}$ is given by:

$$\psi^{n+1} = \psi^n + \Delta t \left[ \left(\frac{3}{2} + \epsilon\right) \mathcal{G}(\psi^n) - \left(\frac{1}{2} + \epsilon\right) \mathcal{G}(\psi^{n-1}) \right], \tag{18}$$

where $\mathcal{G}(\cdot)$ represents the instantaneous tendency from the dynamical core. We set $\epsilon = 0.1$ to damp high-frequency noise while preserving the physical signal. The differentiability of this explicit solver allows gradients to flow through the entire temporal horizon, enabling the neural network to learn corrections that are consistent with the model's time-stepping history.

**Boundary Conditions and Vertical Physics.** Distinct from atmospheric dynamics, ocean circulation is strictly constrained by complex coastlines, where improper handling of the land-sea interface can induce severe numerical instabilities. To mitigate these boundary effects and prevent non-physical noise propagation, we implement a conservative boundary masking strategy (Shu et al., 2025b). We define a binary validity mask $\mathbf{M} \in \{0, 1\}^{D \times H \times W}$ that effectively creates a safety buffer by eroding the coastline. The mask is formalized as:

$$\mathbf{M}(i, j) = \begin{cases} 0, & \text{if cell } (i, j) \text{ is land or has any land neighbor,} \\ 1, & \text{otherwise.} \end{cases} \tag{19}$$

This ensures that all active computational cells are surrounded exclusively by valid ocean points, ensuring stable stencil operations.

**Configuration of HybridOM Variants.** We instantiate two variants of the HybridOM framework with spatial resolutions of $0.5°$ and $0.25°$. The key physical parameters for these configurations are detailed in Table 3.

**Spatial-temporal Filtering for Numerical Stability.** Extended integration of hybrid models often faces stability challenges due to the accumulation of high-frequency noise and spectral blocking. Beyond the intrinsic regularization within the

*Table 3.* **Physical Parameters of HybridOM Variants.** Key configuration settings for the dynamical core at different resolutions. Abbreviations: $\Delta t$ (Time Step), $A_h$ (Horizontal Viscosity), $\mathcal{K}_h$ (Horizontal Diffusivity), $\epsilon$ (AB2 Damping Factor).

| PARAMETER | HYBRIDOM-0.5 | HYBRIDOM-0.25 |
|---|---|---|
| GRID RESOLUTION ($\Delta x$) | $0.5°$ | $0.25°$ |
| GRID TOPOLOGY | ARAKAWA C-GRID | ARAKAWA C-GRID |
| TIME STEP ($\Delta t$) | $7200\,\text{s}$ | $3600\,\text{s}$ |
| HORIZ. VISCOSITY ($A_h$) | $500\,\text{m}^2/\text{s}$ | $500\,\text{m}^2/\text{s}$ |
| HORIZ. DIFFUSIVITY ($\mathcal{K}_h$) | $100\,\text{m}^2/\text{s}$ | $100\,\text{m}^2/\text{s}$ |
| AB2 OFF-CENTERING ($\epsilon$) | $0.1$ | $0.1$ |
| VERTICAL LAYERS | 23 (LAYERED) | 12 (LAYERED) |
| ADVECTION SCHEME | 1ST-ORDER UPWIND | 1ST-ORDER UPWIND |

dynamical core (e.g., upwind schemes), we implement an explicit two-stage stabilization strategy during inference to ensure robust long-term forecasting.

*1) Adaptive Temporal Damping and Clamping.* To prevent "explosive" dynamical kernel's output when the model state drifts from the training distribution (typically after 10 days), we apply a variable-specific damping schedule to the state increments. Let $\tilde{\mathbf{X}}_{t+1}$ be the raw predicted state and $\Delta \mathbf{X} = \tilde{\mathbf{X}}_{t+1} - \mathbf{X}_t$ be the total tendency. We first impose a physical hard constraint via clamping to bound the maximum instantaneous change rate:

$$\Delta \hat{\mathbf{X}} = \text{Clamp}(\Delta \mathbf{X}, -\tau, \tau), \tag{20}$$

where $\tau$ is a threshold (e.g., 0.5 for tracers, 0.2 for velocity).

*2) Dynamic Gaussian Spatial Smoothing.* To counteract grid-scale noise accumulation without suppressing resolved physics in the early phase, we introduce a Dynamic Gaussian Filter. The smoothing intensity, characterized by the standard deviation $\sigma(t)$, ramps up linearly over time:

$$\sigma(t) = \begin{cases} 0 & t < t_{\text{start}} \\ \sigma_{\max} \cdot \frac{t - t_{\text{start}}}{T_{\text{end}} - t_{\text{start}}} & t \geq t_{\text{start}} \end{cases} \tag{21}$$

where $t_{\text{start}} = 10$ days and $\sigma_{\max} = 0.5$. This filter is applied channel-wise: $\mathbf{X}_t \leftarrow G_{\sigma(t)} * \mathbf{X}_t$. This technique is widely employed in numerical simulations, typically serving to stabilize the dynamical core output and suppress non-physical noise (Kochkov et al., 2024).

## C. Details of the Neural Network Architecture

The neural components of HybridOM are designed to capture multi-scale ocean dynamics, balancing the need for local turbulence resolution and global teleconnection modeling. Below, we delineate the detailed architectures of the Ocean Dynamics Model (ODM) and the Flux Gating Model (FGM), providing precise definitions for all mathematical operations and variables.

### C.1. Ocean Dynamics Model (ODM)

The ODM, denoted as $\mathcal{N}_\theta$, serves as the "neural flesh". It adopts a U-shaped hierarchical encoder-decoder structure augmented with a physics-informed latent evolution core.

**1. State Encoder and Decoder.** The input state $\mathbf{X} \in \mathbb{R}^{T \times C \times H \times W}$ represents a spatiotemporal tensor, where $T$ is the number of input time steps, $C$ is the number of physical variables (e.g., $u, v, T, S$), and $H \times W$ denotes the spatial grid resolution. The `StateEncoder` projects this input into a latent feature space via a sequence of `RestrictionBlocks`. Each block performs spatial downsampling:

$$\mathbf{Z}_{enc}^{(l)} = \sigma(\text{GN}(\text{Conv}_{3 \times 3}(\mathbf{Z}_{enc}^{(l-1)}))), \quad s = 2, \tag{22}$$

where $\mathbf{Z}_{enc}^{(l)}$ is the feature map at level $l$, $\text{Conv}_{3 \times 3}$ denotes a 2D convolution with a kernel size of $3 \times 3$ and stride $s = 2$, $\text{GN}(\cdot)$ represents Group Normalization for training stability, and $\sigma(\cdot)$ is the LeakyReLU activation function. A skip connection

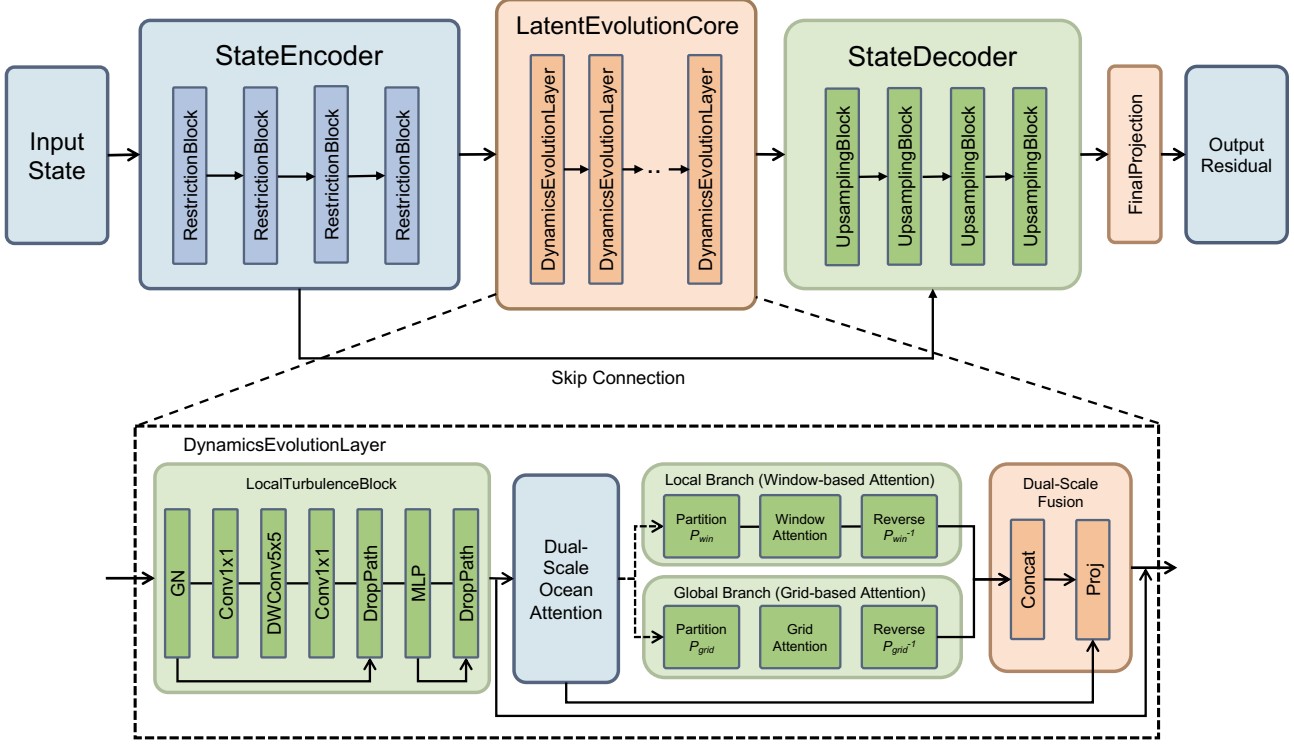

*Figure 7.* Schematic Overview of the HybridOM Neural Architecture of $\mathcal{N}_\theta, \mathcal{N}_{\text{inv}}$.

$\mathbf{Z}_{skip}$ is extracted from the first level output to preserve high-frequency spatial details lost during downsampling. Conversely, the `StateDecoder` reconstructs the residual corrections $\mathcal{R}$ from the evolved latent features using `ConvTranspose2d` (transposed convolution) operations, fusing with $\mathbf{Z}_{skip}$ via channel concatenation prior to the final $1 \times 1$ projection layer.

**2. Latent Evolution Core.** The bottleneck of the ODM is the `LatentEvolutionCore`, which stacks $L$ layers of `DynamicsEvolutionLayer`. To efficiently model fluid dynamics, we design a hybrid block selector that alternates between Convolutional blocks (for local turbulence) and Attention blocks (for global circulation).

*a) Local Turbulence Block.* For modeling small-scale eddies and local mixing, we employ the `LocalTurbulenceBlock`. It follows a pre-norm residual design:

$$\mathbf{Z}' = \mathbf{Z} + \text{DropPath}(\text{Conv}_{1\times1}(\text{DWConv}_{5\times5}(\text{Conv}_{1\times1}(\text{GN}(\mathbf{Z}))))), \tag{23}$$

$$\mathbf{Z}_{out} = \mathbf{Z}' + \text{DropPath}(\text{MLP}(\text{GN}(\mathbf{Z}'))). \tag{24}$$

Here, $\mathbf{Z}$ is the input feature tensor, and GN denotes Group Normalization. The term $\text{DWConv}_{5\times5}$ refers to a $5 \times 5$ depth-wise convolution, utilized to capture local gradients and advective tendencies with expanded receptive fields while minimizing parameter count. MLP denotes a multi-layer perceptron for channel-wise mixing. DropPath is a stochastic depth regularization technique that randomly drops residual branches during training to prevent overfitting.

*b) Dual-Scale Ocean Attention (DSOA).* To effectively model the multi-scale nature of ocean dynamics—ranging from localized eddies to basin-wide teleconnections—we design a dual-branch attention mechanism. Unlike standard global attention which incurs quadratic complexity $\mathcal{O}((HW)^2)$, DSOA decomposes the feature map $\mathbf{Z} \in \mathbb{R}^{H \times W \times C}$ into two parallel subspaces by splitting the attention heads. Let $N_h$ be the total number of heads; the first $N_h/2$ heads are assigned to the Local Branch, and the remaining $N_h/2$ to the Global Branch.

**Local Branch (Window-based Attention).** This branch captures fine-grained, short-range interactions (e.g., sub-mesoscale turbulence). We partition the feature map $\mathbf{Z}$ into non-overlapping windows of size $M \times M$ (with $M = 8$ in our implementation). Mathematically, the partition operator $\mathcal{P}_{win}$ transforms the tensor:

$$\mathcal{P}_{win} : \mathbf{Z} \to \hat{\mathbf{Z}}_{win} \in \mathbb{R}^{(\frac{H}{M} \times \frac{W}{M}) \times (M^2) \times C_{head}}, \tag{25}$$

where the dimensions correspond to the number of windows ($\frac{H}{M} \times \frac{W}{M}$), the sequence length within each window ($M^2$), and the dimension per head ($C_{head} = C/N_h$). The self-attention is computed independently within each window:

$$\mathbf{Y}_{local} = \text{Softmax} \left( \frac{\mathbf{Q}_{win}\mathbf{K}_{win}^T}{\sqrt{d_k}} \right) \mathbf{V}_{win}, \tag{26}$$

where $\mathbf{Q}_{win}, \mathbf{K}_{win}, \mathbf{V}_{win}$ are the query, key, and value projections restricted to the local window. The scaling factor $\sqrt{d_k}$ (where $d_k = C_{head}$) is used to stabilize gradients. This operation limits the receptive field to $M \times M$ but maintains high local resolution.

**Global Branch (Grid-based Attention).** To capture long-range dependencies and global circulation patterns without the computational cost of full attention, we introduce a **Grid Partition** strategy. Instead of grouping adjacent pixels, we group pixels with a fixed stride $M$. The grid partition operator $\mathcal{P}_{grid}$ reshapes and permutes the input tensor to gather spatially sparse but globally distributed tokens:

$$\mathcal{P}_{grid}(\mathbf{Z}) = \text{Reshape}(\mathbf{Z}, [H/M, M, W/M, M, C])$$
$$\xrightarrow{\text{Permute}} \hat{\mathbf{Z}}_{grid} \in \mathbb{R}^{(M^2) \times (\frac{H}{M} \times \frac{W}{M}) \times C_{head}}. \tag{27}$$

Here, the sequence length is $\frac{HW}{M^2}$, representing a coarse-grained global view. For a specific grid index $(i, j)$ in the $M \times M$ mesh, the attention attends to all positions $(y, x)$ in the original map such that $y \equiv i \pmod M$ and $x \equiv j \pmod M$. This effectively creates a dilated attention mechanism that covers the entire spatial domain.

**Dual-Scale Fusion.** The outputs from both branches are reversed to the original spatial layout ($\mathcal{P}^{-1}$) and fused via channel concatenation, followed by a linear projection to mix multi-scale information:

$$\mathbf{Z}_{out} = \text{Proj} \left( \text{Concat} \left[ \mathcal{P}_{win}^{-1}(\mathbf{Y}_{local}), \mathcal{P}_{grid}^{-1}(\mathbf{Y}_{global}) \right] \right) + \mathbf{Z}_{in}. \tag{28}$$

where Proj is a linear projection layer and $\mathbf{Z}_{in}$ is the input tensor added via residual connection. This design allows HybridOM to simultaneously resolve local high-frequency turbulence and global low-frequency waves within a single computational block.

### C.2. Flux Gating Model (FGM)

The Flux Gating Model ($\mathcal{G}_\phi$) is designed to dynamically fuse physical fluxes from the coarse global parent ($\mathbf{F}^l$) and the fine regional child ($\mathbf{F}^h$). As implemented in the `FluxGatingUnit`, the architecture employs a sophisticated two-stage fusion strategy that goes beyond simple linear interpolation.

**Adaptive Soft-Gating (The Selector).** First, a lightweight convolutional `SelectorNet` acts as a dynamic valve. It takes the concatenated flux history from both the fine grid ($\mathbf{F}_t^h$) and the coarse grid ($\mathbf{F}_t^l, \mathbf{F}_{t+1}^l$), along with atmospheric forcing $\mathbf{A}$, to generate spatial attention maps via a Softmax operation. Let $\mathbf{W} \in \mathbb{R}^{3 \times C \times H \times W}$ denote the pixel-wise weights for the fine flux, current coarse flux, and future coarse flux, respectively. The preliminary fused flux $\mathbf{F}_{pre}$ is computed as:

$$\mathbf{F}_{pre} = \mathbf{W}_h \odot \mathbf{F}_t^h + \mathbf{W}_{l,t} \odot \text{Interp}(\mathbf{F}_t^l) + \mathbf{W}_{l,t+1} \odot \text{Interp}(\mathbf{F}_{t+1}^l), \tag{29}$$

where $\odot$ denotes element-wise multiplication, $\text{Interp}(\cdot)$ represents upsampling to match the fine grid resolution, and the weights satisfy the constraint $\sum \mathbf{W} = \mathbf{1}$. This mechanism allows the model to adaptively trust high-resolution physics in turbulent regions (e.g., boundary currents) while stabilizing with coarse trends in the open ocean.

**Deep Residual Refinement (The Refiner).** The linear combination in Stage 1 may still harbor kinematic inconsistencies. To correct this, $\mathbf{F}_{pre}$ is fed into a deep refinement network that mirrors the architecture of the ODM (Sec. 3.3). It consists of

a `StateEncoder`, a `LatentEvolutionCore` (equipped with Dual-Scale Ocean Attention), and a `StateDecoder`. This network predicts a non-linear residual flux correction $\Delta\mathbf{F}$, yielding the final gated flux:

$$\tilde{\mathbf{F}} = \mathbf{F}_{pre} + \text{Decoder}(\text{Core}(\text{Encoder}(\mathbf{F}_{pre} \oplus \mathbf{A}))). \tag{30}$$

where $\oplus$ denotes channel-wise concatenation. This design ensures that the downscaled fluxes are not only consistent with the global boundary conditions but also physically coherent at the fine scale.

## D. Data Details

### D.1. Ocean State Variables and Data Sources

The construction of the ground truth dataset is strictly tailored to the specific requirements of global simulation, regional downscaling, and operational forecasting tasks. We utilize three distinct data products from the Copernicus Marine Service (CMEMS) to construct our training and **GLORYS12V1 Reanalysis**, the operational **GLO12 Analysis**, and the **GLO12 Nowcast**.

**1. Global and Regional Simulation (Hindcast Mode).** For the simulation tasks, our primary source is the CMEMS GLORYS12V1 reanalysis product (Jean-Michel et al., 2021). This dataset provides a consistent, high-resolution ($1/12°$) 3D representation of ocean physics spanning from 1993 to 2020.

- **Global Simulation:** We utilize the entire GLORYS12V1 archive. The data is spatially downsampled to construct two experimental benchmarks: a coarse version at $0.5°$ (HybridOM-0.5) and a high resolution version at $0.25°$ (HybridOM-0.25).

- **Regional Simulation:** We focus on the North Pacific region, defined by the bounding box $120°E - 150°W$ and $15° - 60°N$. The GLORYS12V1 data within this window is downsampled to $0.25°$ to serve as the high-resolution ground truth for boundary-value problem experiments.

**2. Ocean Forecasting (Operational Mode).** For the forecasting task, we adhere to the strict evaluation protocol defined by OceanBench. While the model is pre-trained on the historical GLORYS12V1 data, the testing phase mimics a real-world operational scenario using data from the year 2024:

- **Initial Conditions (IC):** We utilize the **GLO12 Nowcast** product to initialize the model. These snapshots represent the best estimate of the ocean state available in real-time. As indicated in Table 4, Nowcast data is provided at 7-day intervals from Jan 17, 2024, to Dec 28, 2024.

- **Ground Truth (GT):** The model forecasts are evaluated against the **GLO12 Analysis** product. Unlike the Nowcast, the Analysis data incorporates delayed observations to provide a more accurate posterior estimate of the true ocean state.

**Variable Selection and Data Structure.** Across all tasks, the prognostic state vector $\mathbf{X}$ constitutes a high-dimensional representation of the ocean's physical state. It comprises four 3D volumetric variables—Zonal Velocity ($u$), Meridional Velocity ($v$), Salinity ($S$), and Potential Temperature ($T$)—alongside one 2D surface variable, Sea Surface Height ($\eta$).

To balance the need for resolving upper-ocean thermodynamics with computational efficiency, we selectively retain specific vertical layers from the native grid, focusing on the biologically and dynamically active zones extending from the surface down to the permanent thermocline at approximately 644 meters. The vertical discretization strategy is tailored to the model resolution to optimize memory usage:

- **HybridOM-0.5:** We retain the complete set of top 23 discretized depth levels (in meters): 0.49, 2.65, 5.08, 7.93, 11.4, 15.8, 21.6, 29.4, 40.3, 55.8, 77.9, 92.3, 110, 131, 156, 186, 222, 266, 318, 380, 454, 541, and 644 m.

- **HybridOM-0.25:** To accommodate the increased computational load of the eddy-resolving grid, we select a representative subset of 12 layers: 0.49, 5.08, 11.4, 21.6, 40.3, 77.9, 110, 156, 222, 318, 454, and 644 m.

*Table 4.* **Specifications of Source Datasets.** We utilize three CMEMS products. GLORYS12V1 is used for training and historical simulation. For the forecasting task (following OceanBench), GLO12 Nowcast provides the initial conditions, while GLO12 Analysis serves as the validation ground truth. All products share the same native spatial resolution ($1/12°$) and vertical grid (50 levels), from which we select 23 layers.

| DATASET | ROLE | TIME COVERAGE | TEMPORAL FREQ. | NATIVE RES. |
|---|---|---|---|---|
| **GLORYS12V1** | REANALYSIS | 1993-01-01 → 2020-12-31 | DAILY MEAN | $1/12°$ |
| **GLO12 ANALYSIS** | OPERATIONAL GT | 2024-01-01 → 2024-12-31 | DAILY MEAN | $1/12°$ |
| **GLO12 NOWCAST** | INITIAL CONDITION | 2024-01-17 → 2024-12-28 | EVERY 7 DAYS | $1/12°$ |

*Table 5.* **Atmospheric Forcing Configuration.** For simulation tasks, we use ground truth ERA5 snapshots. For forecasting tasks, we generate forcing dynamically using the FuXi-2.0 weather model initialized with ERA5 analysis fields.

| FORCING VARIABLE | TASK | DATA SOURCE | TEMPORAL TYPE | ROLE |
|---|---|---|---|---|
| $u_{10}, v_{10}, t_{2m}$ | SIMULATION | ERA5 REANALYSIS | SNAPSHOT (12:00 UTC) | GROUND TRUTH DRIVER |
| | FORECASTING | FUXI-2.0 MODEL | 10-DAY FORECAST ROLLOUT | PREDICTED DRIVER |
| *FuXi-2.0 Initialization Inputs (Only for Forecasting Task)* | | | | |
| UPPER AIR ($Z, T, U, V, Q$) | FORECASTING | ERA5 ANALYSIS | SNAPSHOT (12:00 UTC) | MODEL INPUT |
| SURFACE ($MSL, U10, \ldots$) | FORECASTING | ERA5 ANALYSIS | SNAPSHOT (12:00 UTC) | MODEL INPUT |

Consequently, the full ocean state is mathematically formalized as a tensor $\mathbf{X} \in \mathbb{R}^{C \times H \times W}$. The total channel dimension $C$ is 93, derived by stacking the multi-level variables (4 vars × 23 levels = 92) with the single-level surface height (1 var). Depending on the spatial resolution, the tensor shapes for our two model variants are: **HybridOM-0.5:** $\mathbf{X} \in \mathbb{R}^{93 \times 360 \times 720}$, corresponding to the $0.5°$ grid; **HybridOM-0.25:** $\mathbf{X} \in \mathbb{R}^{49 \times 720 \times 1440}$.

Detailed specifications of the source datasets, including their temporal coverage and sampling frequencies, are provided in Table 4.

## D.2. Atmospheric Forcing Strategy

The ocean model is driven by atmospheric boundary conditions, specifically 10-meter Zonal Wind ($u_{10}$), 10-meter Meridional Wind ($v_{10}$), and 2-meter Air Temperature ($t_{2m}$). The source of this forcing depends strictly on the nature of the task—whether it is a reanalysis-driven simulation or a realistic forward forecast.

In the **Simulation Tasks** (both Global and Regional), we assume "perfect" atmospheric forcing to isolate the ocean model's errors. We utilize the ERA5 reanalysis dataset, extracting daily snapshots at 12:00 UTC. These snapshots are interpolated from their native $0.25°$ resolution to match the corresponding ocean grid. Conversely, for the **Forecasting Task**, we adopt a realistic operational setting where future atmospheric conditions are unknown. We employ the state-of-the-art **FuXi-2.0** weather forecasting model to generate the forcing trajectory. FuXi-2.0 is initialized with the ERA5 analysis field at 12:00 UTC of the current day. The model consumes a comprehensive set of initial conditions, including upper-air variables (Geopotential $Z$, Temperature $T$, Wind Components $U/V$, Specific Humidity $Q$) across 13 pressure levels, as well as surface variables (MSL, T2M, U10, V10, etc. See (Zhong et al., 2024) for a complete set of variables). FuXi-2.0 then performs a 10-day autoregressive rollout to produce the predicted atmospheric forcing $\hat{\mathbf{A}}_{t:t+10}$, which subsequently drives the HybridOM system. The configuration of atmospheric data is detailed in Table 5.

## D.3. Preprocessing

**Hybrid Normalization Strategy.** We implement a multi-stage preprocessing pipeline to satisfy the distinct requirements of the physical solver and the neural network.

First, to mitigate the dominance of low-frequency seasonal signals, we subtract the climatological mean from variables with strong seasonal periodicity—specifically Temperature ($T$), Salinity ($S$), and Sea Surface Height ($\eta$). This operation acts as a high-pass filter, allowing the model to focus on capturing high-frequency anomalies and dynamic perturbations (Gao et al., 2026).

*Table 6.* **Chronological Data Splitting and Sampling Strategy.** We ensure strict temporal separation between training, validation, and testing sets to evaluate generalization capability.

| PHASE | TIME PERIOD | TASK FOCUS | SAMPLING STRIDE | TOTAL SAMPLES |
|-------|-------------|-----------|-----------------|---------------|
| TRAINING | 1993 – 2018 | MODEL OPTIMIZATION | EVERY 3 DAYS | $\sim 3100$ |
| VALIDATION | 2019 | HYPERPARAM TUNING | EVERY 7 DAYS | $\sim 52$ |
| TEST (SIM) | 2020 (JAN 1 - DEC 31) | LONG-TERM SIMULATION | EVERY 3 DAYS | 80 |
| TEST (FCST) | 2024 (JAN 17 - DEC 28) | 10-DAY FORECASTING | EVERY 7 DAYS | 50 |

Second, we employ a dual-path feeding strategy. For the *physical skeleton*, input variables retain their original physical magnitudes (unnormalized) to ensure the rigorous validity of the primitive equations. Conversely, for the *neural flesh* $\mathcal{N}_\theta$, both the ocean state $\mathbf{X}$ and atmospheric forcing $\mathbf{A}$ are normalized to facilitate stable convergence. Specifically, for any variable $v$, the normalized input $\hat{v}$ is computed as:

$$\hat{v} = \frac{v - \mu_v}{\sigma_v}, \tag{31}$$

where $\mu_v$ and $\sigma_v$ denote the global mean and standard deviation computed from the training set.

**Dataset Splitting.** The dataset spans a period from 1993 to 2024. We strictly partition the data chronologically into training, validation, and testing sets. For the training phase (1993–2018), we sample initial conditions with a stride of 3 days to maximize data diversity while maintaining computational feasibility. The year 2019 is reserved for validation. The testing phase is further divided by task: 2020 is used to evaluate long-term simulation stability, sampled every 3 days (approx. 80 samples). The year 2024 is dedicated to the forecasting task, following the **OceanBench** protocol, where initial conditions are sampled every 7 days from Jan 17th to Dec 28th. This rigorous splitting strategy is summarized in Table 6.

# E. Experimental Details

## E.1. Evaluation Metrics

To rigorously assess the forecasting capability of HybridOM against baselines, we employ five complementary metrics: Weighted RMSE (wRMSE) for global field accuracy, Critical Success Index (CSI) for extreme events, Lagrangian Trajectory Error (LTE), Geostrophic Error alongside Upper Ocean Heat Content (OHC) deviation to verify physical consistency.

**Latitude-Weighted RMSE (Root Mean Square Error).** Standard RMSE is unsuitable for global modeling on regular lat-lon grids because grid cell areas diminish towards the poles. To account for this spherical distortion, we compute the Latitude-Weighted RMSE. Let $\hat{y}_{t,i,j}$ and $y_{t,i,j}$ denote the predicted and ground truth values at time $t$, latitude $i$, and longitude $j$, respectively. The metric is defined as:

$$\text{RMSE}(t) = \sqrt{\frac{\sum_{i,j} w_i (\hat{y}_{t,i,j} - y_{t,i,j})^2}{\sum_{i,j} w_i}}, \tag{32}$$

where $w_i = \cos(\phi_i)$ represents the area weight for latitude $\phi_i$. This ensures that errors in the tropics (large area) contribute proportionally more than errors near the poles.

**Critical Success Index (CSI).** To evaluate the model's performance in forecasting extreme events (e.g., Marine Heatwaves or high-velocity currents), we utilize the Critical Success Index, also known as the Threat Score. For a given threshold $\tau$ (e.g., the 90th percentile of the climatological distribution), we convert the continuous predictions into binary event maps. The CSI is calculated as:

$$\text{CSI} = \frac{\text{Hits}}{\text{Hits} + \text{False Alarms} + \text{Misses}}, \tag{33}$$

where *Hits* represents correctly predicted events, *False Alarms* denotes predicted events that did not occur, and *Misses* indicates observed events that were not predicted. CSI ranges from 0 to 1, with 1 indicating perfect detection of extreme anomalies.

**Lagrangian Trajectory Error.** Ocean currents act as the fundamental transport mechanism for critical substances, including heat, nutrients, and marine debris (e.g., microplastics). Consequently, the Lagrangian trajectories of passive tracers serve as

a powerful proxy for evaluating the local and global dynamical consistency of the simulation, capturing coherent transport structures that purely Eulerian metrics might miss. Formally, the motion of a passive tracer is governed by the ordinary differential equation (ODE):

$$\frac{d\mathbf{x}(t)}{dt} = \mathbf{v}(\mathbf{x}(t), t), \quad \mathbf{x}(t_0) = \mathbf{x}_0 \tag{34}$$

where $\mathbf{x}(t)$ denotes the particle position and $\mathbf{v}$ represents the velocity field. To quantify the transport error, we solve this initial value problem using the explicit **Runge-Kutta 4th Order (RK4)** integration scheme. Given a time step $\Delta t$ (set to 1 hour in our study), the particle position $\mathbf{x}_{n+1}$ at step $n+1$ is updated via:

$$\mathbf{x}_{n+1} = \mathbf{x}_n + \frac{\Delta t}{6}(\mathbf{k}_1 + 2\mathbf{k}_2 + 2\mathbf{k}_3 + \mathbf{k}_4) \tag{35}$$

Here, the slopes $\mathbf{k}_i$ represent velocity estimates at different stages of the interval: $\mathbf{k}_1 = \mathbf{v}(\mathbf{x}_n, t_n)$, $\mathbf{k}_2 = \mathbf{v}(\mathbf{x}_n + \frac{\Delta t}{2}\mathbf{k}_1, t_n + \frac{\Delta t}{2})$, and so forth. Crucially, since the velocity outputs from the model are discrete in both space and time, we implement a dual interpolation strategy within the integration loop. For spatial query points off the grid, we apply bilinear interpolation using the four nearest neighbors; for temporal queries between forecast frames (e.g., at $t + \Delta t/2$), we apply linear interpolation between the instantaneous fields at day $t$ and $t + 1$. We initialize a uniform $10 \times 10$ grid of tracers in the central region of the domain and compute the mean Euclidean distance between the predicted trajectories and the ground truth trajectories after a 10-day integration period. We selected four representative regions to evaluate the Lagrangian Trajectory Error, as shown in Table 7: the East Pacific (EP), Gulf Stream (GULF), Southern Ocean (SO), and Southern Atlantic (SA).

*Table 7.* **Definition of Evaluation Regions.** The geographical bounding boxes (Longitude and Latitude) for the four key regions used in Lagrangian trajectory analysis, derived from the $1/12°$ global grid indices.

| REGION NAME | ABBR. | LONGITUDE RANGE | LATITUDE RANGE |
|---|---|---|---|
| EAST PACIFIC | EP | $160°\text{W} - 120°\text{W}$ | $30°\text{N} - 50°\text{N}$ |
| GULF STREAM | GULF | $70°\text{W} - 50°\text{W}$ | $25°\text{N} - 45°\text{N}$ |
| SOUTHERN OCEAN | SO | $180°\text{W} - 140°\text{W}$ | $63°\text{S} - 23°\text{S}$ |
| SOUTHERN ATLANTIC | SA | $45°\text{W} - 25°\text{W}$ | $40°\text{S} - 20°\text{S}$ |

**Geostrophic Balance** We assess the dynamical consistency between the predicted Sea Surface Height ($\eta$) and surface currents by calculating the geostrophic velocities. The geostrophic currents ($u_g, v_g$) are derived from the gradients of $\eta$ according to the geostrophic balance relation:

$$u_g = -\frac{g}{f}\frac{\partial \eta}{\partial y}, \quad v_g = \frac{g}{f}\frac{\partial \eta}{\partial x}, \tag{36}$$

where $g$ is the gravitational acceleration and $f = 2\Omega \sin\phi$ is the Coriolis parameter at latitude $\phi$. We compute the Geostrophic Error by comparing the derived velocities from the predicted height $\hat{\eta}$ against those derived from the ground truth height $\eta_{gt}$:

$$\text{ERROR}_{geo} = \sqrt{\frac{\sum_{i,j} w_{i,j} \left[(u_g(\hat{\eta}) - u_g(\eta_{gt}))^2 + (v_g(\hat{\eta}) - v_g(\eta_{gt}))^2\right]}{\sum_{i,j} w_{i,j}}}, \tag{37}$$

where $w_{i,j} = \cos(\phi_{i,j})$ represents the latitude-dependent area weights.

**Upper Ocean Heat Content (OHC)** Ocean Heat Content is a critical indicator of the ocean's thermal energy storage. We calculate the OHC by vertically integrating the potential temperature $T$ from the surface down to a reference depth $D$ (approx. 644m, covering the upper 23 layers):

$$\text{OHC} = \int_D^0 \rho_0 C_p T(z)\, dz \approx \rho_0 C_p \sum_{k=1}^{K} T_k \Delta z_k, \tag{38}$$

where $\rho_0 = 1025\,\text{kg m}^{-3}$ is the reference seawater density, $C_p = 3996\,\text{J kg}^{-1}{}^\circ\text{C}^{-1}$ is the specific heat capacity, and $\Delta z_k$ is the thickness of the $k$-th vertical layer. The prediction accuracy is evaluated using the spatially weighted RMSE of the 2D OHC field:

$$\text{ERROR}_{\text{OHC}} = \sqrt{\sum_{i,j}(\text{OHC}_{i,j}^{\text{pred}} - \text{OHC}_{i,j}^{\text{gt}})^2}. \tag{39}$$

## E.2. Model Training

**Implementation and Hardware.** All models, including HybridOM and baselines, are implemented using the PyTorch framework. Experiments are conducted on a high-performance computing cluster equipped with 8 NVIDIA A100 GPUs (80GB). To maximize computational efficiency, we employ the Distributed Data Parallel (DDP) strategy, distributing the workload across all available devices.

**Optimization and Hyperparameters.** Unless otherwise specified, we maintain a consistent training configuration across all experiments to ensure fair comparison:

- **Batch Size:** We set the batch size to 1 per GPU, resulting in an effective global batch size of 8. This configuration is chosen to accommodate the high memory footprint of the high-resolution ocean state tensors and the gradients required for the differentiable solver.

- **Optimizer:** We utilize the **Adam** optimizer with default parameters ($\beta_1 = 0.9, \beta_2 = 0.999$) for gradient descent.

- **Learning Rate Schedule:** To stabilize convergence, we employ a Cosine Annealing schedule ('CosineAnnealingLR'). The learning rate is initialized at $\eta_{max} = 1 \times 10^{-3}$ and strictly decays following a cosine curve to a minimum of $\eta_{min} = 1 \times 10^{-8}$ over the course of training.

- **Training Duration:** All models are trained for a total of 100 epochs. Early stopping is applied if the validation loss does not improve for 10 consecutive epochs to prevent overfitting.

## E.3. Model Parameters

We provide the detailed hyperparameter configurations for the HybridOM components and all baseline models used in our experiments. Table 8-9 details the specific architectures of the two neural components in the Ocean Dynamics Model (ODM): the state residual network $\mathcal{N}_\theta$ and the inverse-Laplacian operator $\mathcal{N}_{inv}$. Table 10 and Table 11 summarize the key parameters for the AI-for-Earth baselines and the general Computer Vision / Spatio-Temporal baselines, respectively.

*Table 8.* **HybridOM-0.5 Architecture Configurations.** We utilize distinct configurations for the state evolution and the inverse Laplacian operator.

| HYPERPARAMETER | $\mathcal{N}_\theta$ | $\mathcal{N}_{INV}$ |
|---|---|---|
| EMBEDDING DIMENSION | 256 | 256 |
| LATENT DIMENSION | 512 | 512 |
| SPATIAL DOWNSCALING LAYERS | 4 | 6 |
| TEMPORAL EVOLUTION LAYERS | 8 | 4 |
| INPUT RESOLUTION | $360 \times 720$ | $360 \times 720$ |

*Table 9.* **HybridOM-0.25 Architecture Configurations.** We utilize distinct configurations for the state evolution and the inverse Laplacian operator.

| HYPERPARAMETER | $\mathcal{N}_\theta$ | $\mathcal{N}_{INV}$ |
|---|---|---|
| EMBEDDING DIMENSION | 256 | 128 |
| LATENT DIMENSION | 512 | 256 |
| SPATIAL DOWNSCALING LAYERS | 6 | 6 |
| TEMPORAL EVOLUTION LAYERS | 6 | 2 |
| INPUT RESOLUTION | $720 \times 1440$ | $720 \times 1440$ |

## E.4. Baseline Implementations

To ensure a rigorous evaluation, we benchmark HybridOM against the baseline models listed in Table 10 and 11. Our implementation strategy is divided into two categories:

*Table 10.* **AI-for-Earth (AI4E) Baseline Configurations.** Key parameters for weather and climate forecasting models.

| MODEL | IN/OUT CH. | EMBED DIM | DEPTH/LAYERS | HEADS | WINDOW/GRID |
|---|---|---|---|---|---|
| FENGWU | 99 / 93 | 256 | 8 | 8 | $8 \times 8$ |
| PANGU-WEATHER | 99 / 93 | 192 | [2, 6, 6, 2] | [6, 12, 12, 6] | [2, 6, 12] |
| FUXI | 99 / 93 | 256 | 48 (U-TRANS) | - | - |
| FOURCASTNET | 99 / 93 | 256 | 12 | 16 | - |
| CIRT | 99 / 93 | 768 | 12 | 12 | - |
| GRAPHCAST | 99 / 93 | 512 | 16 (GNN) | - | MESH NODES |
| ONEFORECAST | 99 / 93 | 512 | 16 (GNN) | - | MESH NODES |

*Table 11.* **General CV & Spatio-Temporal Baseline Configurations.** Standard architectures adapted for ocean forecasting.

| MODEL | IN/OUT CH. | HIDDEN DIM | LAYERS (DEPTH) | SPECIFIC PARAMS |
|---|---|---|---|---|
| CONVLSTM | 99 / 93 | 64 | 3 | KERNEL: $3 \times 3$ |
| SIMVP | 99 / 93 | $S$: 128, $T$: 256 | $N_S$: 4, $N_T$: 8 | TRANSLATOR: INCEPTION |
| PASTNET | 99 / 93 | 256 | $N_T$: 8 | CODEBOOK: 128 |
| DIT (DIFF. TRANS.) | 99 / 93 | 128 | $N_S$: 4, $N_T$: 8 | PATCH SIZE: 2 |
| RESNET | 99 / 93 | 64 | - | DROPOUT: 0.1 |
| U-NET | 99 / 93 | 64 (BASE) | 4 (LEVELS) | SKIP CONNECTIONS |
| NNG | 99 / 93 | 128 | - | LOCAL CNN |

**Official Pre-trained Models.** For the domain-specific foundation models, **NeuralOM** (Gao et al., 2026) and **WenHai** (Cui et al., 2025), we utilize their officially released checkpoints for direct inference. This ensures the comparison reflects their optimal reported performance.

**Models Trained from Scratch.** For the remaining general spatiotemporal forecasting architectures, we train them from scratch using the same dataset and evaluation protocol as HybridOM. Detailed hyperparameter configurations for these baselines are provided in Tables 7–10 in the Appendix.

**Unified Formulation.** Consistent with our framework, all baselines $\mathcal{N}_{\text{baseline}}$ trained from scratch are formulated as an autoregressive predictor:

$$\hat{\mathbf{X}}_{t+1} = \mathcal{N}_{\text{baseline}}(\mathbf{X}_t, \mathbf{A}_{t:t+1}). \tag{40}$$

Specifically, the current ocean state $\mathbf{X}_t \in \mathbb{R}^{93 \times H \times W}$ is channel-wise concatenated with the atmospheric forcing sequence $\mathbf{A}_{t:t+1} \in \mathbb{R}^{6 \times H \times W}$, forming a unified input tensor $\in \mathbb{R}^{99 \times H \times W}$.

## F. Limitations and Future Work

While HybridOM demonstrates promising results in balancing physical consistency with computational efficiency, several limitations remain to be addressed in future research.

First, although our hybrid paradigm significantly reduces the computational cost compared to traditional numerical solvers during inference, the training process still demands substantial GPU memory, particularly for high-resolution settings. Future work will explore advanced engineering strategies to scale up HybridOM to full-depth, kilometer-scale simulations.

Second, a natural progression for our research is to extend HybridOM into a fully coupled ocean-atmosphere system, allowing for two-way feedback mechanisms. Furthermore, incorporating a prognostic sea ice module—rather than relying on prescribed boundary conditions—would significantly enhance the model's fidelity, particularly for climate simulations in polar regions.

*Table 12.* **Efficiency and Complexity Benchmark.** Corrected wall-clock time, model size, FLOPs, and memory usage measured with batch size 1 on a single NVIDIA A100 GPU (80G).

| Model | Train ms | Infer ms | Params | Infer FLOPs | Train Mem (MB) | Infer Mem (MB) |
|-------|----------|----------|--------|-------------|----------------|----------------|
| HybridOM-05 | 782.26 | 510.76 | 66.71M | 3371.01G | 16293.68 | 2824.99 |
| UNet | 98.28 | 35.08 | 6.00M | 267.36G | 5243.00 | 3637.32 |
| ConvLSTM | 66.73 | 22.29 | 0.97M | 503.74G | 2911.60 | 1977.96 |
| CirT | 227.84 | 73.80 | 253.08M | 180.44G | 7640.48 | 3345.24 |
| FourCastNet | 242.32 | 47.48 | 28.44M | 378.36G | 7813.68 | 2433.40 |
| PastNet | 315.30 | 116.02 | 19.85M | 503.14G | 3879.82 | 842.17 |
| SimVP | 455.66 | 242.22 | 15.05M | 723.87G | 5000.91 | 1341.00 |
| ResNet | 859.10 | 287.86 | 21.59M | 11183.56G | 26555.01 | 2562.30 |
| DiT | 958.94 | 295.55 | 11.11M | 314.48G | 5405.24 | 1676.64 |
| GraphCast | 1444.36 | 484.78 | 35.32M | 7180.25G | 45956.69 | 10548.46 |
| NeuralOM | 2162.22 | 675.11 | 110.20M | 8382.13G | 62688.61 | 8401.26 |
| OneForecast | 2338.96 | 689.63 | 50.02M | 6174.13G | 51292.34 | 9702.98 |

*Table 13.* **Runtime Against POM Numerical Solvers.** One-day runtime on a single NVIDIA A100 GPU under the same benchmark setup.

| Model | One-day Runtime | Relative to HybridOM-0.5 |
|-------|-----------------|--------------------------|
| HybridOM-0.5 | 510.76 ms | 1.00× |
| POM-explicit | 107871.89 ms | 211.20× |
| POM-implicit | 138525.83 ms | 271.21× |

# G. Additional Results

## G.1. Additional Evaluations

### G.1.1. INFERENCE EFFICIENCY

We report the training and inference costs of HybridOM against neural baselines in Table 12. This benchmark uses batch size 1, one-step next-day supervised training, one-step rollout inference, and a single NVIDIA A100 GPU (80G). The measurements show that HybridOM-0.5 has higher per-step cost than compact purely neural models, but remains cheaper than the strong global ocean/weather forecasting baselines.

To contextualize the inference cost against conventional ocean numerical solvers, Table 13 compares HybridOM-0.5 with GPU-accelerated POM using explicit and implicit numerical schemes. POM, the Princeton Ocean Model, is a classical physics-based ocean circulation model that solves three-dimensional primitive equations for ocean thermodynamics and dynamics (Xu et al., 2015). The POM setup uses longitude $0°$–$360°$, latitude $80°$N–$80°$S, $0.5°$ horizontal resolution, 24 vertical interfaces, 23 active levels, a 300s integral interval, and 288 time steps per day.

### G.1.2. WEIGHTED MSE OF GLOBAL SIMULATIONS

Table 14 presents the forecasting performance (weighted MSE) of HybridOM across lead times ranging from 10 to 60 days, benchmarked against various state-of-the-art baselines. To ensure a fair comparison, all models were trained and evaluated at a unified resolution of $0.5°$.

Figure 9 reports an additional spectral analysis. The spectra provide a complementary view to point-wise errors by assessing whether the model preserves energy across spatial scales.

### G.1.3. INITIAL-CONDITION SENSITIVITY

We evaluate whether HybridOM's rollout response is dominated by the ocean initial condition or by perturbation-scale-dependent numerical amplification. In this diagnostic, only the initial ocean state is perturbed; atmospheric forcing and ocean-land boundary treatment are kept fixed. We use HybridOM-0.5 over a 10-day horizon and evaluate four representative

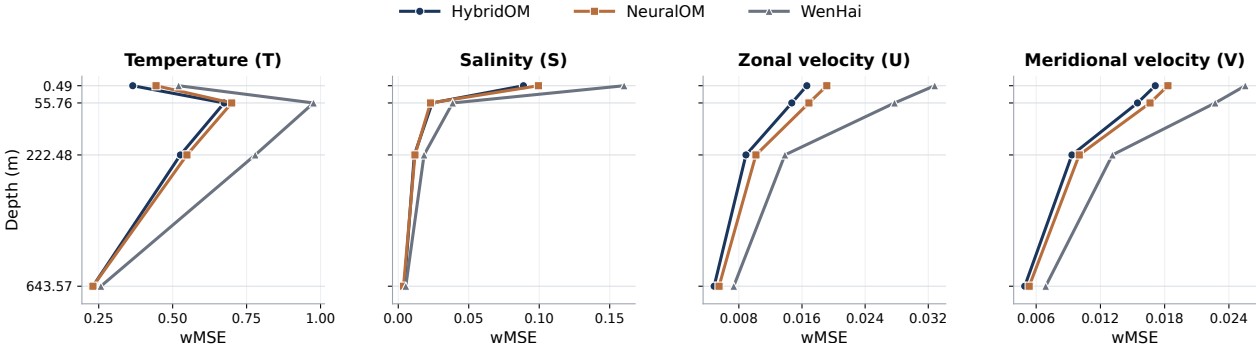

*Figure 8.* **Representative-depth wMSE.** Weighted MSE as a function of depth at the 40-day lead for temperature, salinity, zonal velocity, and meridional velocity. Lower values indicate better long-term simulation accuracy.

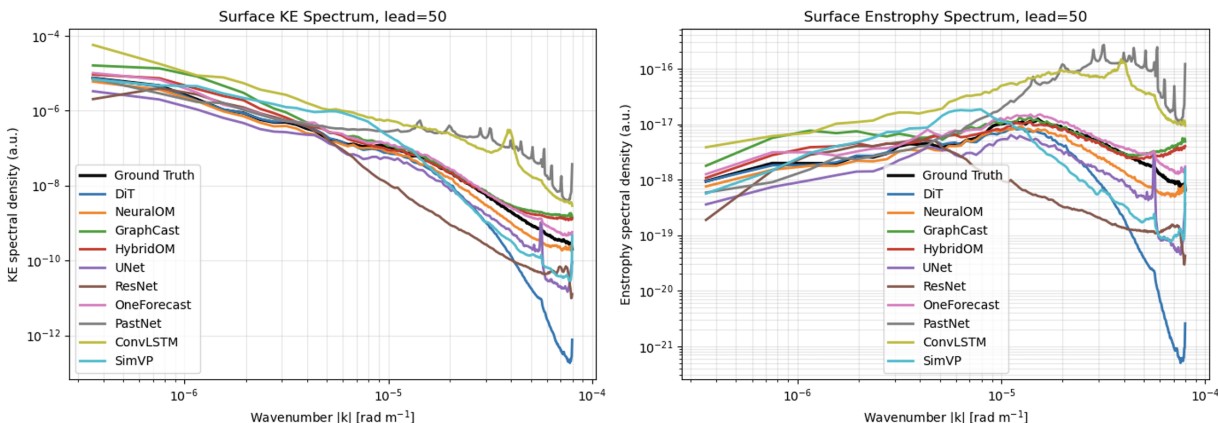

*Figure 9.* **Spectral Analysis.** Energy spectra comparison used to assess whether HybridOM preserves physically meaningful multi-scale structures.

start days. For each start day, we construct a perturbed initial ocean state

$$\mathbf{x}_0' = \mathbf{x}_0 + \delta\mathbf{x}_0, \qquad \delta\mathbf{x}_0 = \varepsilon_{\mathrm{IC}}(\sigma_T\xi_T, \sigma_S\xi_S, \sigma_u\xi_u, \sigma_v\xi_v, \sigma_\eta\xi_\eta), \tag{41}$$

where $\xi \sim \mathcal{N}(0,1)$ is standard normal noise applied to the corresponding prognostic fields and $(\sigma_T, \sigma_S, \sigma_u, \sigma_v, \sigma_\eta) = (0.5828, 0.3208, 0.1473, 0.1123, 0.0643)$ are variable-wise standard deviations. We test two perturbation amplitudes, $\varepsilon_{\mathrm{IC}} = 0.01$ and $\varepsilon_{\mathrm{IC}} = 0.05$.

At lead time $\tau$, the normalized finite-difference sensitivity is computed as

$$S(\tau) = \frac{\|F_\tau(\mathbf{x}_0', \mathbf{A}) - F_\tau(\mathbf{x}_0, \mathbf{A})\|_{\mathrm{norm}}}{\|\delta\mathbf{x}_0\|_{\mathrm{norm}}}, \tag{42}$$

where $F_\tau$ denotes the HybridOM rollout under the same atmospheric forcing $\mathbf{A}$, and $\|\cdot\|_{\mathrm{norm}}$ is the root-mean-square over state variables after normalizing each variable by its standard deviation.

As shown in Table 15, the close agreement between the two perturbation amplitudes indicates that HybridOM is sensitive to the ocean initial state in the expected dynamical sense, while the response remains approximately local and stable over this perturbation range.

### G.1.4. WEIGHTED RMSE OF GLOBAL FORECASTING

Table 16 illustrates the forecasting performance (weighted RMSE) of our two HybridOM variants at different resolutions across lead times of 1, 3, 5, 7, and 9 days, benchmarked against the OceanBench baselines (El Aouni et al., 2026). As

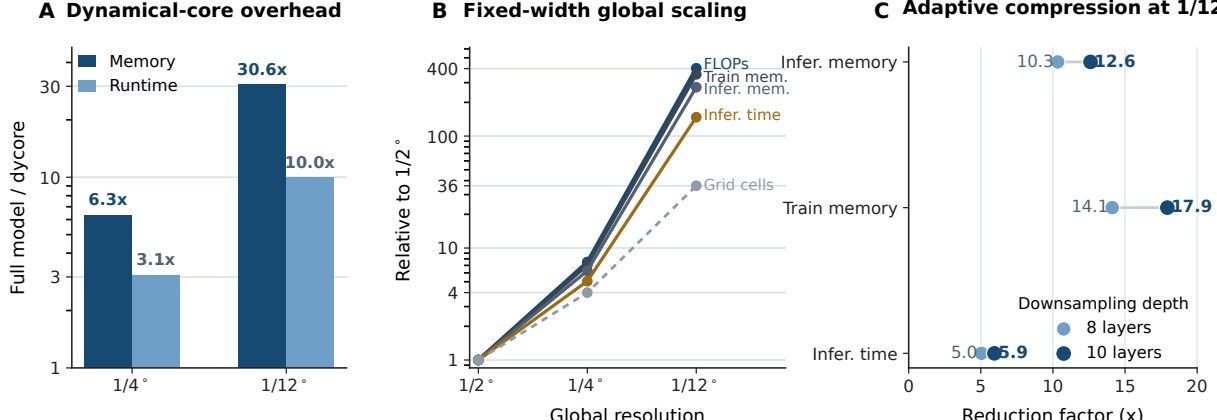

*Figure 10.* **High-resolution scaling and adaptive compression.** (A) Full-model inference memory and runtime relative to the dynamical core alone at $1/4°$ and $1/12°$. (B) Fixed-width global scaling normalized to the $1/2°$ setting, compared with grid-cell scaling. (C) Reduction factors from adaptive compression at $1/12°$ relative to the naive full-model configuration. The two compression settings use 8 or 10 spatial downsampling layers in both $\mathcal{N}_\theta$ and $\mathcal{N}_{\text{inv}}$ (In our naive setup, $\mathcal{N}_\theta$ has 4 spatial layers while $\mathcal{N}_{\text{inv}}$ has 6 layers).

observed, our models achieve state-of-the-art (SOTA) results on the majority of metrics, with the exception of ocean current velocities at greater depths (450m). **To ensure a fair comparison, the forecast outputs of all participating models were uniformly downsampled to a resolution of** $1/12°$ **and evaluated against the** $1/12°$ **GLO12 Analysis data.** Notably, the HybridOM-0.25 model, leveraging its higher resolution, excels in short-term forecasting ($< 5$ days), whereas HybridOM-0.5 demonstrates superior long-term stability.

### G.1.5. HIGH-RESOLUTION SCALING AND ADAPTIVE COMPRESSION

Figure 10 summarizes how HybridOM scales toward higher-resolution global modeling. First, the differentiable dynamical core itself is relatively lightweight: at $1/4°$, the full model uses $6.3\times$ inference memory and $3.1\times$ runtime compared with the core alone, and at $1/12°$ these ratios increase to $30.6\times$ and $10.0\times$. Thus, the neural components dominate the additional cost, and their share becomes larger after scaling up; the physical core is not the main computational bottleneck. Second, naively preserving the original fixed-width neural configuration at $1/12°$ leads to a very large resource demand, reaching 771.1 GB inference memory, 5800.6 GB training memory, and 75.2 s one-step inference time. Third, this cost can be substantially reduced by increasing the spatial downsampling depth of both the state residual network $\mathcal{N}_\theta$ and the inverse operator $\mathcal{N}_{\text{inv}}$. With 10 downsampling layers in both modules, adaptive compression reduces inference memory, training memory, and inference time by $12.6\times$, $17.9\times$, and $5.9\times$, respectively, relative to the naive $1/12°$ full configuration.

### G.2. Additional Visualizations of Global Simulations

In this section, we provide a representative visualization of the global ocean simulation generated by HybridOM. Figure 11 illustrates Sea Surface Temperature (SST), Sea Surface Salinity (SSS), Surface Velocity (VEL), and Sea Surface Height (SSH) for a rollout initialized on January 1, 2020. This visualization demonstrates the model's capability to maintain stable, energetic, and physically realistic ocean dynamics without numerical drift.

### G.3. Additional Visualizations of Global Forecasting

In this section, we present a representative global ocean forecasting visualization. Figure 12 displays the forecasted fields side-by-side with the ground truth (GLO12 analysis) for a rollout initialized on January 1, 2024. For better visualization and to highlight the dynamic evolution, the initial conditions have been subtracted from all displayed results. The close agreement between the forecast and the analysis data highlights the model's strong predictive skill in capturing both large-scale circulation patterns and local anomalies.

*Table 14.* **Long-term Forecast Accuracy (wMSE).** Comparison of HybridOM and baselines across lead times of 10-60 days. Best results are **bolded**, second best underlined. Values exceeding 10 are denoted as −.

| MODEL | 10D | 20D | 30D | 40D | 50D | 60D |
|---|---|---|---|---|---|---|
| **TEMPERATURE (T)** | | | | | | |
| HYBRIDOM | **0.168** | **0.307** | **0.414** | **0.496** | **0.562** | **0.614** |
| NEURALOM | 0.193 | 0.344 | 0.447 | 0.528 | 0.597 | 0.656 |
| ONEFORECAST | 0.217 | 0.438 | 0.658 | 0.913 | 1.199 | 1.500 |
| GRAPHCAST | 0.217 | 0.437 | 0.648 | 0.882 | 1.133 | 1.378 |
| DIT | 0.206 | 0.433 | 0.628 | 0.811 | 1.017 | 1.341 |
| FOURCASTNET | 0.255 | 0.567 | - | - | - | - |
| PASTNET | 0.471 | 2.811 | 5.924 | 7.607 | 8.257 | 8.540 |
| SIMVP | 0.223 | 1.277 | 1.377 | 1.397 | 1.422 | 1.452 |
| UNET | 0.254 | 0.441 | 0.563 | 0.655 | 0.731 | 0.797 |
| CONVLSTM | 0.312 | 0.681 | 1.095 | 1.806 | 2.774 | 3.576 |
| CIRT | 0.624 | 0.783 | 0.860 | 0.905 | 0.936 | 0.958 |
| RESNET | 0.373 | 0.651 | 0.768 | 0.824 | 0.865 | 0.897 |
| WENHAI | 0.170 | 0.382 | 0.563 | 0.724 | 0.870 | 1.006 |
| **SALINITY (S)** | | | | | | |
| HYBRIDOM | **0.011** | **0.019** | **0.025** | **0.029** | **0.032** | **0.035** |
| NEURALOM | 0.012 | 0.021 | 0.026 | 0.031 | 0.034 | 0.037 |
| ONEFORECAST | 0.014 | 0.031 | 0.059 | 0.099 | 0.155 | 0.225 |
| GRAPHCAST | 0.015 | 0.031 | 0.054 | 0.086 | 0.122 | 0.157 |
| DIT | 0.014 | 0.027 | 0.041 | 0.060 | 0.086 | 0.122 |
| FOURCASTNET | 0.018 | 0.036 | - | - | - | - |
| PASTNET | 0.037 | 0.283 | 0.642 | 0.843 | 0.934 | 0.975 |
| SIMVP | 0.016 | 0.085 | 0.090 | 0.091 | 0.092 | 0.094 |
| UNET | 0.017 | 0.028 | 0.034 | 0.039 | 0.043 | 0.047 |
| CONVLSTM | 0.025 | 0.058 | 0.105 | 0.200 | 0.333 | 0.444 |
| CIRT | 0.045 | 0.051 | 0.054 | 0.055 | 0.056 | 0.057 |
| RESNET | 0.027 | 0.044 | 0.054 | 0.061 | 0.066 | 0.070 |
| WENHAI | 0.013 | 0.026 | 0.038 | 0.049 | 0.059 | 0.070 |
| **ZONAL VELOCITY (U)** | | | | | | |
| HYBRIDOM | **0.005** | **0.008** | **0.010** | **0.012** | **0.014** | **0.015** |
| NEURALOM | 0.005 | 0.010 | 0.012 | 0.014 | 0.015 | 0.016 |
| ONEFORECAST | 0.006 | 0.010 | 0.015 | 0.021 | 0.028 | 0.035 |
| GRAPHCAST | 0.006 | 0.011 | 0.016 | 0.022 | 0.032 | 0.044 |
| DIT | 0.005 | 0.011 | 0.016 | 0.021 | 0.030 | 0.050 |
| FOURCASTNET | 0.008 | 0.019 | - | - | - | - |
| PASTNET | 0.014 | 0.071 | 0.132 | 0.157 | 0.164 | 0.167 |
| SIMVP | 0.006 | 0.044 | 0.047 | 0.049 | 0.051 | 0.053 |
| UNET | 0.008 | 0.013 | 0.016 | 0.018 | 0.020 | 0.022 |
| CONVLSTM | 0.010 | 0.020 | 0.034 | 0.062 | 0.097 | 0.123 |
| CIRT | 0.018 | 0.020 | 0.021 | 0.022 | 0.022 | 0.023 |
| RESNET | 0.010 | 0.017 | 0.021 | 0.023 | 0.025 | 0.026 |
| WENHAI | 0.006 | 0.012 | 0.018 | 0.022 | 0.026 | 0.028 |
| **MERIDIONAL VELOCITY (V)** | | | | | | |
| HYBRIDOM | **0.005** | **0.008** | **0.011** | **0.013** | **0.014** | **0.015** |
| NEURALOM | 0.005 | 0.009 | 0.012 | 0.014 | 0.015 | 0.016 |
| ONEFORECAST | 0.006 | 0.011 | 0.015 | 0.019 | 0.024 | 0.028 |
| GRAPHCAST | 0.006 | 0.011 | 0.015 | 0.021 | 0.028 | 0.034 |
| DIT | 0.006 | 0.012 | 0.016 | 0.020 | 0.025 | 0.032 |
| FOURCASTNET | 0.008 | 0.016 | - | - | - | - |
| PASTNET | 0.013 | 0.063 | 0.121 | 0.148 | 0.156 | 0.159 |
| SIMVP | 0.007 | 0.037 | 0.040 | 0.042 | 0.043 | 0.045 |
| UNET | 0.007 | 0.012 | 0.014 | 0.016 | 0.016 | 0.017 |
| CONVLSTM | 0.009 | 0.017 | 0.029 | 0.058 | 0.099 | 0.129 |
| CIRT | 0.013 | 0.015 | 0.016 | 0.016 | 0.017 | 0.017 |
| RESNET | 0.008 | 0.013 | 0.014 | 0.015 | 0.015 | 0.016 |
| WENHAI | 0.005 | 0.011 | 0.015 | 0.018 | 0.020 | 0.022 |

*Table 15.* **Initial-condition sensitivity.** Normalized finite-difference response of HybridOM-0.5 under ocean-state perturbations.

| Perturbation | Day 1 | Day 5 | Day 10 |
|---|---|---|---|
| $\epsilon = 0.01$ | 0.9486 | 1.0432 | 1.2551 |
| $\epsilon = 0.05$ | 0.9473 | 1.0404 | 1.2486 |

*Table 16.* **Global Ocean Forecasting wRMSE.** Comparison of HybridOM variants ($0.5°$ and $0.25°$) against SOTA baselines (GLONet, WenHai, XiHe). The table reports the weighted RMSE for Potential Temperature ($T$), Salinity ($S$), Zonal Velocity ($U$), and Meridional Velocity ($V$) across three depths: Surface, 220m, and 450m. Results are reported for forecast lead times of 1, 3, 5, 7, and 9 days. Best results are highlighted in **bold**; second-best are underlined.

| MODEL | DEPTH: SURFACE | | | | DEPTH: 220M | | | | DEPTH: 450M | | | |
|---|---|---|---|---|---|---|---|---|---|---|---|---|
| | $T$ | $S$ | $U$ | $V$ | $T$ | $S$ | $U$ | $V$ | $T$ | $S$ | $U$ | $V$ |
| *Lead Time: 1 Day* | | | | | | | | | | | | |
| GLONET | 0.451 | 0.197 | 0.106 | 0.095 | 0.467 | 0.070 | 0.071 | 0.057 | 0.546 | 0.058 | 0.063 | 0.049 |
| WENHAI | 0.418 | 0.218 | 0.128 | 0.120 | 0.327 | 0.061 | 0.051 | 0.053 | 0.249 | 0.046 | 0.044 | 0.046 |
| XIHE | 0.511 | 0.316 | 0.098 | 0.098 | 0.445 | 0.079 | 0.051 | 0.052 | 0.329 | 0.060 | **0.043** | **0.045** |
| HYBRIDOM ($0.5°$) | 0.271 | 0.160 | 0.079 | 0.080 | 0.310 | 0.057 | 0.052 | 0.053 | **0.226** | **0.044** | 0.044 | **0.045** |
| HYBRIDOM ($0.25°$) | **0.250** | **0.155** | **0.071** | **0.071** | **0.292** | **0.053** | **0.049** | **0.050** | 0.247 | 0.045 | 0.047 | 0.049 |
| *Lead Time: 3 Days* | | | | | | | | | | | | |
| GLONET | 0.515 | 0.235 | 0.113 | 0.100 | 0.523 | 0.082 | 0.074 | 0.060 | 0.441 | 0.064 | 0.064 | **0.052** |
| WENHAI | 0.615 | 0.276 | 0.146 | 0.132 | 0.409 | 0.076 | 0.063 | 0.064 | **0.314** | **0.058** | 0.054 | 0.056 |
| XIHE | 0.567 | 0.321 | 0.108 | 0.106 | 0.509 | 0.088 | **0.061** | **0.062** | 0.375 | 0.066 | **0.052** | 0.054 |
| HYBRIDOM ($0.5°$) | 0.359 | 0.211 | 0.100 | 0.099 | 0.421 | 0.078 | 0.070 | 0.070 | 0.317 | 0.059 | 0.061 | 0.061 |
| HYBRIDOM ($0.25°$) | **0.338** | **0.210** | **0.091** | **0.090** | **0.397** | **0.073** | 0.066 | 0.065 | 0.326 | 0.059 | 0.061 | 0.062 |
| *Lead Time: 5 Days* | | | | | | | | | | | | |
| GLONET | 0.617 | 0.269 | 0.126 | 0.115 | 0.584 | 0.094 | 0.079 | **0.067** | 0.454 | 0.071 | 0.068 | **0.058** |
| WENHAI | 0.803 | 0.320 | 0.165 | 0.143 | 0.495 | 0.089 | 0.077 | 0.077 | 0.524 | 0.068 | 0.066 | 0.067 |
| XIHE | 0.627 | 0.353 | 0.121 | 0.118 | 0.545 | 0.093 | **0.071** | 0.071 | 0.548 | 0.071 | **0.060** | 0.062 |
| HYBRIDOM ($0.5°$) | **0.375** | **0.180** | 0.106 | 0.104 | 0.459 | 0.082 | 0.076 | 0.076 | **0.352** | **0.061** | 0.067 | 0.067 |
| HYBRIDOM ($0.25°$) | **0.375** | 0.182 | **0.104** | **0.102** | 0.458 | **0.081** | 0.076 | 0.075 | 0.362 | 0.063 | 0.069 | 0.069 |
| *Lead Time: 7 Days* | | | | | | | | | | | | |
| GLONET | 0.705 | 0.302 | 0.141 | 0.132 | 0.655 | 0.106 | 0.086 | **0.077** | 0.495 | 0.078 | 0.074 | **0.067** |
| WENHAI | 0.981 | 0.353 | 0.182 | 0.150 | 0.565 | 0.100 | 0.087 | 0.087 | 0.431 | 0.076 | 0.075 | 0.076 |
| XIHE | 0.676 | 0.373 | 0.130 | 0.126 | 0.597 | 0.101 | **0.078** | 0.080 | 0.448 | 0.075 | **0.068** | 0.070 |
| HYBRIDOM ($0.5°$) | **0.431** | **0.203** | **0.120** | **0.117** | 0.524 | 0.092 | 0.085 | 0.085 | **0.404** | **0.068** | 0.075 | 0.076 |
| HYBRIDOM ($0.25°$) | 0.439 | 0.207 | **0.120** | **0.117** | 0.531 | **0.092** | 0.086 | 0.086 | 0.420 | 0.071 | 0.077 | 0.078 |
| *Lead Time: 9 Days* | | | | | | | | | | | | |
| GLONET | 0.802 | 0.327 | 0.154 | 0.143 | 0.715 | 0.114 | 0.092 | **0.085** | 0.532 | 0.084 | 0.079 | **0.073** |
| WENHAI | 1.147 | 0.381 | 0.195 | 0.158 | 0.620 | 0.109 | 0.095 | 0.095 | 0.470 | 0.082 | 0.083 | 0.083 |
| XIHE | 0.723 | 0.402 | 0.136 | 0.131 | 0.611 | 0.104 | **0.084** | **0.085** | 0.459 | 0.078 | **0.073** | 0.074 |
| HYBRIDOM ($0.5°$) | **0.471** | **0.217** | **0.129** | **0.125** | **0.560** | 0.098 | 0.090 | 0.090 | **0.435** | **0.073** | 0.080 | 0.081 |
| HYBRIDOM ($0.25°$) | 0.487 | 0.223 | 0.130 | 0.126 | 0.574 | **0.098** | 0.092 | 0.093 | 0.457 | 0.076 | 0.083 | 0.084 |

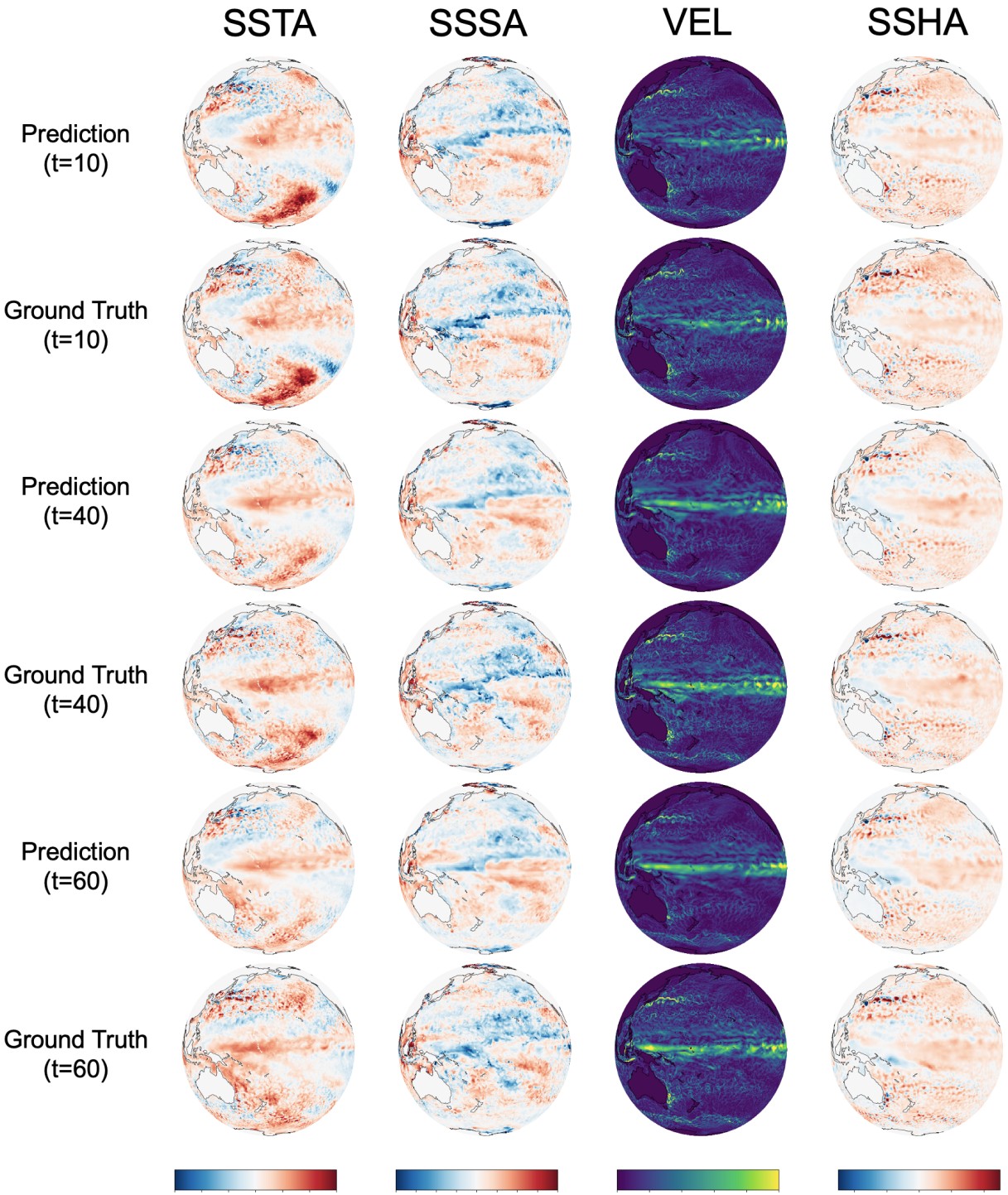

*Figure 11.* **Global Simulation Snapshot.** The simulation is initialized on January 1, 2020.

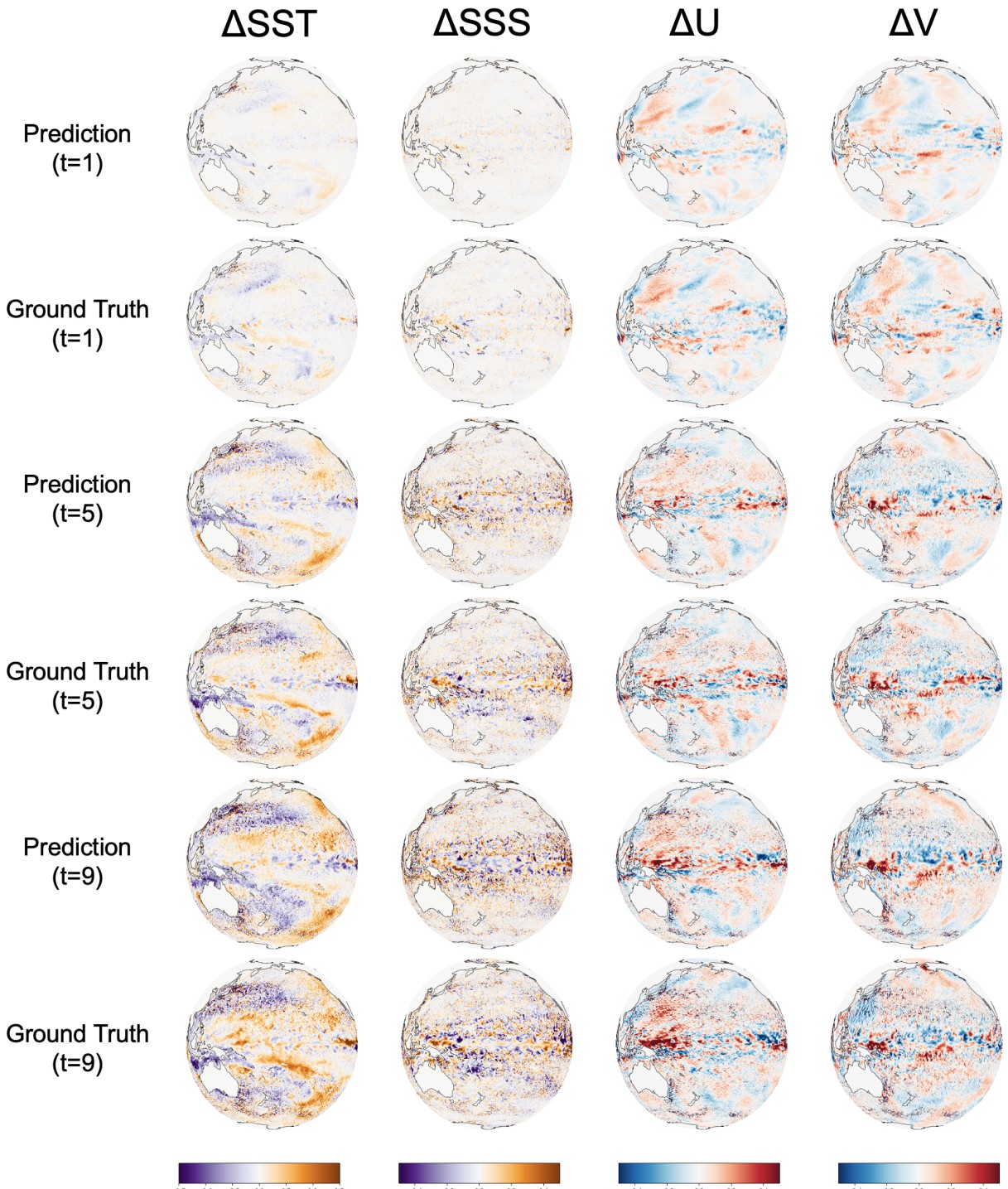

*Figure 12.* **Global Forecasting Snapshot.** The simulation is initialized on January 1, 2024.

