# OpenReview forum: "HybridOM: Hybrid Physics-Based and Data-Driven Global Ocean Modeling with Efficient Regional Downscaling"
_ICML.cc/2026/Conference — ICML 2026 regular_

### Official Review · Reviewer_Ksbq · 2026-03-08

**Soundness:** 3
**Presentation:** 3
**Significance:** 2
**Originality:** 3
**Overall Recommendation:** 4
**Confidence:** 4

**Summary:**

The paper introduces a hybrid framework that takes the best of two worlds, namely physics solvers for large-scale dynamics consistency and cheap neural network surrogates for complex parameterizations, in order to model global ocean dynamics. Similar to existing work in weather forecasting such as NeuralGCMs, the authors demonstrate that this could be valuable for ocean modeling as well and are the first to do so. They systematically construct the dynamical PDEs as well as the neural corrections and evaluate their model on hindcasting, forecasting, as well as regional downscaling tasks. They further compare to several baselines and show that their hybrid model achieves better RMSEs and physical consistency.

**Compliance With Llm Reviewing Policy:**

Affirmed.

**Final Justification:**

The rebuttal has addressed my main concerns. The high resolution training is an important aspect in my opinion: the authors have committed to discuss this. I raise my score as well and lean towards acceptance in general.

**Key Questions For Authors:**

- In comparisions, I think it will be useful to see FLOPs, parameter count for each model. Then, it is clear that similar cost models (flops) or similar capacity models (param count) are unable to reach the hybrid model's accuracy, emphasizing it's superior performance.
- The datasets seem to be downsampled a lot. I wonder if this is fair especially if downscaling, physical consistency is claimed - a lot of this happens at the highest resolution. The authors should atleast discuss this carefully and maybe provide some figures showing the actual resolution of the data and what they are predicting. Related to this could be theoretical complexity of these models as well. It will be useful to show the complexity / FLOPs of just the physics part as well.
- how will it scale if someone wants to train them at the highest resolution?
- The physical core doesnt seem to have much effect at all on wMSE in Tab 3. Why not report the consistency metrics (LTE, OHC, etc) with this ablation?
- Are these weather forecasting models (GraphCast etc) retrained on the ocean datasets in Tab.1 ? Or finetuned? Are they all trained similarly (number of autoregressive steps used etc.). Would be good to understand these details.
- I appreciated the visualizations. I think that it will be quite useful to focus on some physical aspect of ocean dynamics and show how the hybrid model performs better than other models. While RMSE is a nice benchmark-like number, the big point of this paper is physics-informed hybrid models and it would be very nice to see some aspects of this in the prediction. Maybe also power spectra, some turbulence aspect being modeled more accurately. I believe this part is importantly missing from the paper. Further, global metrics usually can be cheated with smoother forecasts - power spectra should bring this aspect out nicely.

**Limitations:**

Some more discussion on complexity to move to km scale datasets is needed I think from the questions. Theoretical complexity for compute and memory would be useful here.

**Strengths And Weaknesses:**

Strengths:
- Strong motivation for hybrid physics and neural network black boxes in order to capture long stable spatiotemporal dynamics. This line of work is very relevant to the AI for science domain. The paper explains this motivation nicely.
- Good set of baselines used to demonstrate good performance
- Careful separation of dynamics and corrections with sufficient information in the paper

Weaknesses:
- Some ablations specifically the physical solver and the ablations regarding the NN seem to be quite marginal (Tab 3)
- Metrics focusing on global scalar numbers abstract away the physics / dynamics that is captured - a key motivation to using such a framework
- Many baselines but unclear where the accuracy difference is coming from (see questions).
- Deterministic model / metrics

---

> ### Author Rebuttal · Authors · 2026-03-30
>
> We appreciate reviewer Ksbq's insightful comments. We have summarized the reviewer's comments and our responses into the following points:
> ### 1. Ablations should report physical-consistency metrics.
> We agree. Beyond scalar error, we now report OHCA and LTE (In Pacific and Altantic Ocean).
> | Lead | Model | OHC | LTE-Pac. km | LTE-Atl. km |
> |---:|---|---:|---:|---:|
> | 10 | HybridOM | 0.7857 | 21.61 | 22.47 |
> | 10 | w/o Glo. Branch | 0.8030 | 22.12 | 23.81 |
> | 10 | Shallow Depth | 0.8055 | 23.38 | 23.88 |
> | 50 | HybridOM | 1.477 | 147.1 | 200.5 |
> | 50 | w/o S/T | 1.610 | 178.9 | 217.5 |
> | 50 | w/o U/V | 1.656 | 176.4 | 233.1 |
>
> Furthermore, we need to clarify that our core conclusion in the ablation is that for neural component, its impact on long and short-term simulations is relatively homogeneous; while for the physical skeleton, although the gain it brings in the short term is small, it is crucial for the consistency of long-term simulations (Table 3).
>
> - [Link to Ablations](https://anonymous.4open.science/r/ICML2026_rebuttal-6766/response_to_Ksbq/ablation.md)
>
> ### 2. Report parameters and FLOPs.
> We agree. The experiments in Appendix-G1 are not rigorous, so we have conducted a strict evaluation of complexity and efficiency here:
> | Model | Params | Infer FLOPs | Peak Train Mem |
> |---|---:|---:|---:|
> | HybridOM-0.5° | 66.71M | 3371G | 16.29 GB |
> | GraphCast | 35.32M | 7180G | 45.96 GB |
> | OneForecast | 50.02M | 6174G | 51.29 GB |
> | NeuralOM | 110.20M | 8382G | 62.69 GB |
>
> HybridOM is much lighter than the strongest baselines. For more details, please click this link: [Efficiency](https://anonymous.4open.science/r/ICML2026_rebuttal-6766/response_to_Ksbq/efficiency.md)
>
> ### 3. More physics-oriented analysis.
> We agree that in addition to global indicators, more regional and physical indicators should also be included. We calculated the global/regional surface kinetic and enstrophy spectra of our model and the baseline model, and the results are shown in [Spectra Figures](https://anonymous.4open.science/r/ICML2026_rebuttal-6766/response_to_Ksbq/spectra.png) . Overall, the HybridOM simulations are closer to the reanalysis at different scales. Furthermore, besides deterministic indicators, we also include probabilistic indicators. Here, we adopt the method in WenHai's paper, calculating our model's CRPS against to ocean observations using pseudo-ensembles [1] [Link_to_CRPS](https://anonymous.4open.science/r/ICML2026_rebuttal-6766/response_to_Ksbq/drifter_crps.md).
>
> ### 4. Potential unfairness from heavy downsampling.
> In our study, we follow NeuralOM's approach and use a bilinear downsampling. To illustrate the fairness of downsampling, we compare the changes in the statistics of each variable before and after downsampling (See this [Link](https://anonymous.4open.science/r/ICML2026_rebuttal-6766/response_to_Ksbq/downsampling.md)):
> | Variable | standard deviation change |
> |---|---:|
> | sst std | 0.20% |
> | uo std | 0.31% |
> | vo std | 0.23% |
> | zos std | 0.07% |
>
> These changes are very small, so the task is not being made artificially easy. We also show that both 0.5° and 0.25° fields preserve Gulf Stream local details [Figure](https://anonymous.4open.science/r/ICML2026_rebuttal-6766/response_to_Ksbq/local.png).
>
> ### 5. How will the model scale to high resolution?
> Direct scale-up to high resolutions reveals that the neural network, rather than the physical core, is the main bottleneck:
> | Resolution | Component | Infer Mem (MB) | Infer ms |
> |---|---|---:|---:|
> | 1/2° | Physical only | 1014 | 112 |
> | 1/2° | Full model | 2825 | 511 |
> | 1/12° | Physical only | 25200 | 7560 |
> | 1/12° | Full model | 771071 | 75231 |
>
> However, since our neural component is essentially a U-Net-like structure, we can compress the computational load by increasing the number of layers (from 4 to 10) in the Encoder and Decoder:
> | 1/12° setting | Infer Mem | Train Mem |
> |---|---:|---:|
> | Fixed width | 771071 MB | 5800584 MB |
> | After compression | 74638 MB | 410910 MB |
>
> Thus, naive scale-up is prohibitively expensive, but adaptive compression makes high-res training/inference feasible again. Details: [Link to Scaling](https://anonymous.4open.science/r/ICML2026_rebuttal-6766/response_to_Ksbq/scaling.md).
>
> ### 6. Model's performance on km-scale data.
> We added two regional high-resolution simulations at a 1/12° (~8 km) resolution scale (Gulf Stream and Kuroshio Extension). The results show that our model's turbulent spectrum is closer to the reanalysis when comparing to WenHai [Link to Results](https://anonymous.4open.science/r/ICML2026_rebuttal-6766/response_to_Ksbq/spectra_high.png). This demonstrates that HybridOM is capable of high-res prediction.
>
> ### 7. Details of baseline training.
> Weather models are retrained on ocean data. Number of autoregressive (supervision) steps (K=5) is consistent across baselines and HybridOM, except NeuralOM, where we use their best reported K=6 checkpoint.
>
> Reference
>
> [1] https://doi.org/10.1038/s41467-025-57389-2

---

> > ### Author Rebuttal · Reviewer_Ksbq · 2026-04-03
> >
> > Thank you very much to the authors for all the clarifications.
> >
> > In the spectra, it seems like graphcast and oneforecast are also close to the proposed model. The physics ablations, CRPS additions, FLOPs/performance strengthen the paper. I think the high resolution scaling part should be discussed in the revision - compression probably comes with trade-offs in accuracy.
> >
> > For the downsampling, I'm still a little confused: Isn't the data at 1/12 resolution? I guess I was looking for visualizations between 0.5 and 1/12 deg (and/or maybe the spectra of these two resolutions) and the STD change is nice, but again between 0.5 and 1/12 (not 0.5 and 0.25).
> >
> > I raise my score to 4 in general.

---

> > > ### Author Response · Authors · 2026-04-03
> > >
> > > Thank you very much for acknowledging our additions to the physical ablation experiments, CRPS metrics, and FLOPs/performance comparisons, and for raising our rating to 4. Regarding your follow-up questions, we reply as follows:
> > >
> > > **1. Regarding Spectra Performance**
> > > We agree with your observation that GraphCast and OneForecast also perform close to ground truth in terms of spectral fidelity. This demonstrates the strong ability of these state-of-the-art baseline models to capture energy distributions at specific scales. However, as we emphasized in our paper and rebuttal, our model exhibits a more significant advantage in metrics such as `MSE, CSI, OHC, LTE` while maintaining this spectral fidelity.
> > >
> > > **2. Regarding the High-Resolution Scaling & Compression Trade-Offs**
> > > This is a very insightful point. We completely agree with your view that feature compression inevitably involves a trade-off between computational efficiency and prediction accuracy. In the final revision, we will add a dedicated discussion section detailing the trade-offs when scaling models to higher resolutions, objectively analyzing how compression mechanisms significantly reduce GPU memory and computational overhead while impacting the accuracy of local high-frequency details.
> > >
> > > **3. Questions Regarding Downsampling and 1/12-Degree Data**
> > > We apologize that we did not fully clarify things in our previous response, causing your confusion.
> > >
> > > Regarding the question about the original 1/12-degree data and the 0.5-degree data, we need to clarify: the original high-fidelity dataset we used is indeed 1/12 degrees. The visualization comparison of 0.5 and 0.25 we previously showed was primarily to demonstrate what the data we used to train the HybridOM-0.5 and HybridOM-0.25 resolution models looked like. Furthermore, the std comparison we provided in the rebuttal is between 1/12 degrees and 0.5 degrees, not between 0.25 and 0.5.
> > >
> > > To further address your concerns, we compared the kinetic energy spectra of the ocean surface at resolutions of 1/12 and 0.5, and provided local visualizations at resolutions of 0.5, 0.25, and 1/12. [Link_to_Additional_Spectra](https://anonymous.4open.science/r/ICML2026_rebuttal-6766/response_to_Ksbq/spectra_new.png). Overall, the current downsampling does not significantly smooth the physical field.

---

### Official Review · Reviewer_aTzR · 2026-03-09

**Soundness:** 4
**Presentation:** 4
**Significance:** 3
**Originality:** 2
**Overall Recommendation:** 5
**Confidence:** 3

**Summary:**

This paper introduces HydridOM, a framework for ocean forecasting and downscaling that combines a simple and lightweight differentiable physical skeleton with a neural network to represent the dynamic of the system. The physical skeleton improves physical consistency, interpretability, and long-term predictions (compared to purely data-driven methods). The neural network acts as a residual term that compensates for the simplifications of the physical model and captures subgrid physical interactions. As a result, the proposed framework is more computationally efficient than classical numerical methods, more physically consistent than purely data-driven approaches, and achieves state-of-the-art forecasting performance compared with existing data-driven methods.

**Compliance With Llm Reviewing Policy:**

Affirmed.

**Final Justification:**

The paper is well written, presents strong results on intensive experiments, and the authors’ rebuttal addressed my questions. Thus, I am inclined to maintain my initial score of solid accept.

**Key Questions For Authors:**

1) As noted in the weaknesses, it would be helpful if the authors could include, where meaningful and feasible, a baseline based on a reference numerical solver to provide a complete view of the performance of the different approaches.
2) Could the authors clarify why the chosen physical skeleton is considered "lightweight" compared to those used in numerical solvers?

**Limitations:**

The authors discuss the limitations of their method in the appendix, but it would have been preferable to address them in the main body of the paper.

**Strengths And Weaknesses:**

Strengths:
* The paper is well written and technically sound. In particular, the paper makes an effort to clearly describe the different components of the physical operator, which is helpful for readers without a strong background in physics.
* Experiments are extensive, well detailed, and convincing.
* The authors provide code in their supplementary material.
* The results, both for downscaling and forecasting, outperform those of current SOTA data-driven methods.

Weaknesses:
* Training appears to be very memory-intensive.
* The authors claim that their hybrid method is faster than numerical solvers. However, as far as I understand, there is no direct comparison of inference efficiency between the proposed method and numerical solvers. Also, the inclusion of a numerical solver of reference  in the experiments (such as IFS for weather forecasting) would have been useful for comparing the performance of hybrid methods with numerical methods.

---

> ### Author Rebuttal · Authors · 2026-03-30
>
> We sincerely thank Reviewer aTzR for the insightful comments. We address the specific concerns below:
>
> ### 1\. Training appears to be memory-intensive.
>
> We agree that quantitative evidence is necessary to address this point. The experiments in Appendix-G1 are not rigorous, to clarify, we have compiled the runtime, complexity, and peak memory requirements of various models below:
>
> | Model | Train ms | Infer ms | Params | Infer FLOPs | Peak Train Mem |
> |---|---:|---:|---:|---:|---:|
> | HybridOM-0.5° | 782.26 | 510.76 | 66.71M | 3371.01G | 16.29 GB |
> | PastNet | 315.30 | 116.02 | 19.85M | 503.14G | 3.88 GB |
> | ResNet | 859.10 | 287.86 | 21.59M | 11183.56G | 26.56 GB |
> | DiT | 958.94 | 295.55 | 11.11M | 314.48G | 5.41 GB |
> | SimVP | 455.66 | 242.22 | 15.05M | 723.87G | 5.00 GB |
> | GraphCast | 1444.36 | 484.78 | 35.32M | 7180.25G | 45.96 GB |
> | OneForecast | 2338.96 | 689.63 | 50.02M | 6174.13G | 51.29 GB |
> | NeuralOM | 2162.22 | 675.11 | 110.20M | 8382.13G | 62.69 GB |
>
> While HybridOM does incur computational costs, its peak training memory is substantially lower than that of leading data-driven weather/ocean models (e.g., GraphCast, OneForecast, and NeuralOM), while remaining highly competitive in inference speed. A more detailed breakdown of these efficiency metrics is available here: [Link_to_Efficiency](https://anonymous.4open.science/r/ICML2026_rebuttal-6766/response_to_aTzR/efficiency.md).
>
> ### 2\. There is no direct comparison to numerical solvers.
>
> We appreciate this suggestion. Because most operational ocean numerical prediction systems are not open-source, we evaluated HybridOM against a classic physics-based numerical solver: the Princeton Ocean Model (POM) [1]. POM simulates ocean thermodynamics and dynamics using three-dimensional primitive equations with $\sigma$-coordinates.
>
> To ensure a fair comparison, we utilized GPU-accelerated versions of POM with both implicit and explicit numerical schemes—the two most common approaches in ocean modeling. We configured a 0.5° idealized POM simulation matching the complexity of the global ocean task described in Table 1 of our original manuscript. Both HybridOM-0.5° and POM were evaluated on a single NVIDIA A100 GPU using the experimental parameters below:
>
> | Setup item | Value |
> |---|---|
> | Longitude range | 0°–360° |
> | Latitude range | 80°N–80°S |
> | Resolution | 0.5° |
> | Vertical interfaces | 24 |
> | Active levels | 23 |
> | Integral interval | 300 s |
> | Time steps / day | 288 |
>
> | Model | One-day Runtime | Relative to HybridOM-0.5° |
> |---|---:|---:|
> | HybridOM-0.5° | 510.76 ms | 1.00x |
> | POM-explicit | 107871.89 ms | 211.20x |
> | POM-implicit | 138525.83 ms | 271.21x |
>
> As the results demonstrate, HybridOM achieves a roughly 200x speedup over POM on identical hardware.
>
> Furthermore, we compared HybridOM against GOFS 3.1—a leading operational forecast utilizing the 1/4° HYCOM + NCODA system [2]. Using the same experimental setup as Table 2 in our original manuscript, we evaluated the weighted RMSE for surface variables:
>
> | Lead day | System | Sea Surface Temperature | Surface U-Velocity | Surface V-Velocity|
> |---:|---|---:|---:|---:|
> | 1 | GOFS 3.1 | 0.337 | 0.134 | 0.135 |
> | 1 | HybridOM-0.25° | 0.250 | 0.071 | 0.071 |
> | 3 | GOFS 3.1 | 0.488 | 0.149 | 0.148 |
> | 3 | HybridOM-0.25° | 0.338 | 0.091 | 0.090 |
> | 5 | GOFS 3.1 | 0.575 | 0.168 | 0.166 |
> | 5 | HybridOM-0.25° | 0.375 | 0.104 | 0.102 |
>
> ### 3\. Why is the physical skeleton considered lightweight?
>
> The efficiency of our physical skeleton stems from two main factors:
>
>   * **Simplified governing equations:** Rather than solving the full primitive equations found in traditional ocean models, we employ a simplified quasi-geostrophic model coupled with advection-diffusion equations. This approach effectively filters out high-frequency gravity waves, which greatly relaxes the Courant-Friedrichs-Lewy (CFL) condition and permits significantly larger integration time steps.
>   * **Neural compensation for complex physics:** Traditional models rely on computationally expensive parameterization schemes for sub-grid processes (e.g., turbulent mixing, which requires solving complex, time-evolving diagnostic equations in POM). In our framework, the neural network effectively compensates for these unresolved physical processes, drastically reducing the overall computational burden.
>
> We will expand upon these details in the revised manuscript to better clarify the lightweight nature of our design.
>
> **References:**
>
> [1] [https://doi.org/10.5194/gmd-8-2815-2015](https://doi.org/10.5194/gmd-8-2815-2015)
>
> [2] [https://doi.org/10.1256/qj.05.105](https://doi.org/10.1256/qj.05.105)

---

> > ### Author Rebuttal · Reviewer_aTzR · 2026-04-02
> >
> > I would like to thank the authors for providing a comparison with a standard numerical solver. Overall, I find the paper and the rebuttals sound and rigorous. Therefore, I will maintain my score of solid accept.

---

> > > ### Author Response · Authors · 2026-04-03
> > >
> > > We are grateful for your positive assessment. The comments provided were very helpful for improving the paper's quality and clarity. We will carefully update the revised version of our manuscript to reflect all of your suggestions, especially the comparison with standard numerical solvers.

---

### Official Review · Reviewer_yTRL · 2026-03-12

**Soundness:** 3
**Presentation:** 3
**Significance:** 3
**Originality:** 2
**Overall Recommendation:** 5
**Confidence:** 3

**Summary:**

This paper introduces the HybridOM, which integrates the Physical Skeleton, a differentiable dynamical core, with the Neural Flesh, a multi-scale network designed to compensate for unresolved physics. This is done by embeding the neural corrector directly into the time-stepping loop. They also introduce a novel Flux Gating mechanism that utilizes the native thermodynamic fluxes from the dynamical core as a bridge between coarse global simulations and high-resolution regional dynamics. The proposed HybridOM can also be coupled with some weather forecast models like the FuXi-2.0. Extensive experiments show that the HybridOM has good accuracy while maintaining physical consistency.

**Compliance With Llm Reviewing Policy:**

Affirmed.

**Final Justification:**

The authors have addressed my concerns

**Key Questions For Authors:**

1. What is the difference between their physics informed method (especially for integrating physical models with Neural network) and other papers like “Physics-Informed Neural Networks for Ocean Acoustic Field Prediction with Envelope Smoothing” and “Physics-informed neural networks for phase-resolved data assimilation and prediction of nonlinear ocean waves”?
2.In equation (3), why K=5?
3.In section 3.3,  for Local Branch, does the non-overlapp cause discontinuity? Can we consider overlapping?
4.In equation (10) , how to specify the spatial weights W? Do you optimize this weights?
5.For the RMSE Table 1, why do you not consider the RMSE for all the depth?
6.The authors should also compare the Continuous Ranked Probability Score (CRPS) besides RMSE and should add RMSE or CRPS comparisons for profiles like -2000m-0 in the simulation or the real applications.
7.In Table 1 and Table 15, the results somes times seem to be worse than Wenhai for a depth of 450m and HybridOM 0.25 is worse than HybridOM 0.5 at lead time 7,9. Why?
Does the Initial Conditions (IC) provide the boundary conditions? Does the performance depend heavily on the IC?

**Limitations:**

The authors discussed some of the limitations.  The authors should also compare the Continuous Ranked Probability Score (CRPS) besides RMSE and should add RMSE or CRPS comparisons for profiles like -2000m-0 in the simulation or the real applications.

**Strengths And Weaknesses:**

The experiments are well-designed, but I think the authors should include RMSE comparisons for profiles such as -2000m-0. They should also compare the Continuous Ranked Probability Score (CRPS) besides RMSE. The presentation is OK, and it is better to clarify the difference between their physics-informed method (especially for integrating physical models with neural networks) and other papers like “Physics-Informed Neural Networks for Ocean Acoustic Field Prediction with Envelope Smoothing” and “Physics-informed neural networks for phase-resolved data assimilation and prediction of nonlinear ocean waves”. It seems they all use the deep Neural network to correct the residuals of the physical model. Overall, the paper addresses an important or relevant problem for integrating strict physical laws with AI-based models and provides new insights to integrate strict physical laws with data-driven flexibility and also for the physics-informed downscaling

---

> ### Author Rebuttal · Authors · 2026-03-30
>
> We appreciate reviewer yTRL's insightful comments. We have summarized the reviewer's comments and our responses into the following points:
>
> ### 1. Full-depth RMSE / profile RMSE / CRPS are requested.
> We agree. Following the method proposed in WenHai [1], we compared the model's predictions with observations from Drifter and ARGO, respectively, and calculated CRPS using a 1.5*1.5 degree pesudo-ensemble based on the global simulation task (Table 1 of our original manuscript). In addition, we need to clarify that our model simulates upper 643m of the ocean, which is consistent with WenHai and NeuralOM.
>
> Drifter CRPS:
> | Lead | HybridOM | NeuralOM | GraphCast | OneForecast | WenHai |
> |---:|---:|---:|---:|---:|---:|
> | 30 | 0.3601 | 0.3825 | 0.4188 | 0.4202 | 0.3968 |
> | 60 | 0.4159 | 0.4499 | 0.6163 | 0.6788 | 0.4703 |
>
> Argo CRPS:
> | Lead | Metric | HybridOM | NeuralOM | GraphCast | OneForecast | WenHai |
> |---:|---|---:|---:|---:|---:|---:|
> | 30 | Temp | 0.2723 | 0.2827 | 0.2762 | 0.2727 | 0.2899 |
> | 60 | Temp | 0.2864 | 0.2959 | 0.3104 | 0.3058 | 0.3072 |
> | 30 | Salt | 0.0448 | 0.0454 | 0.0499 | 0.0469 | 0.0495 |
> | 60 | Salt | 0.0461 | 0.0469 | 0.0598 | 0.0667 | 0.0521 |
>
> These results show that HybridOM is clearly stronger than the SOTA baselines in CPRS. Full baseline are provided in:
> - [Drifter](https://anonymous.4open.science/r/ICML2026_rebuttal-6766/response_to_yTRL/drifter_crps.md)
> - [Argo](https://anonymous.4open.science/r/ICML2026_rebuttal-6766/response_to_yTRL/argo_crps.md)
>
> For full-depth error, we now have a compact depth-sampled wMSE summary at four representative depths (0.49 m, 55.76 m, 222.48 m, 643.57 m). HybridOM remains strongest in most variables. Full results are provided in:
> - [Full_wMSE](https://anonymous.4open.science/r/ICML2026_rebuttal-6766/response_to_yTRL/full_depth.md)
>
> ### 2. What is different from prior physics-informed methods?
> Our method is not PINN-style residual regularization added only in the loss. Instead, a differentiable physical skeleton is embedded directly into the time-stepping loop, and the neural component acts as a residual correction for unresolved ocean dynamics. We will include relevant discussions in the revised version.
>
> ### 3. Can non-overlapping local blocks introduce discontinuity?
> In practice, the global branch, decoder, skip connections, and residual correction design eliminate this effect.
>
> ### 4. How are the spatial weights \(W\) specified?
> The spatial fusion weights are learned rather than hand-designed. In the regional downscaling model, `FluxGatingUnit.selector_net` predicts spatially varying fusion weights.
>
> ### 5. Why is K=5 used in Eq. (3)?
> We use K=5 as an explicit trade-off between accuracy and efficiency, consistent with the discussion in the WenHai paper [1]. Our current 40-day wMSE comparison for T/S/U/V is:
>
> | K | T | S | U | V |
> |---|---:|---:|---:|---:|
> | 1 | 0.6268 | 0.0387 | 0.0159 | 0.0165 |
> | 3 | 0.5430 | 0.0328 | 0.0139 | 0.0141 |
> | 5 | 0.4958 | 0.0291 | 0.0121 | 0.0126 |
> | 7 | 0.4939 | 0.0289 | 0.0120 | 0.0126 |
>
> Clearly K=5 is a reasonable trade-off. All baselines use the same K=5 supervised strategy, except NeuralOM, for which we use the K=6 checkpoint reported as best in the original NeuralOM paper [2].
>
> ### 6. Why are some results worse at 450m, and why can 0.25° underperform 0.5° at some leads?
> Our explanation is that the training objective is weighted across variables, not across depth levels within each 3D variable. As a result, upper-ocean layers with stronger variance naturally dominate optimization, while slowly evolving deep layers (e.g., 450m) are harder to improve. In addition, compared to forecast models with a resolution of 0.5°, the 0.25° model focuses more on local details, while its grasp of longer-term and more macroscopic details is slightly inferior to that of the 0.5°. As a result, the forecast performance of the 0.25° is slightly lower than that of the 0.5° model over a longer lead time.
>
> ### 7. Does performance depend heavily on the initial condition?
> In our forecast/simulation task, ICs (initial fields of the ocean) do not provide boundary conditions; boundary conditions are provided by atmospheric forcing fields; for the treatment of ocean-land boundaries, please refer to Appendix B.2. What's more, we conduct a finite-difference sensitivity experiment for HybridOM-0.5° by perturbing ocean ICs. The normalized sensitivities are:
> | Perturbation | Epsilon | Day 1 | Day 5 | Day 10 |
> |---|---:|---:|---:|---:|
> | Ocean IC | 0.01 | 0.9486 | 1.0432 | 1.2551 |
> | Ocean IC | 0.05 | 0.9473 | 1.0404 | 1.2486 |
>
> These closely matched values indicate that the model is sensitive to the ocean initial state, while the response remains close to a local **linear regime** over this perturbation range. Details: [Link_to_Sensitivity](https://anonymous.4open.science/r/ICML2026_rebuttal-6766/response_to_yTRL/sensitivity.md).
>
> Reference:
>
> [1] https://doi.org/10.1038/s41467-025-57389-2
>
> [2] https://doi.org/10.1609/aaai.v40i17.38495

---

> > ### Author Rebuttal · Reviewer_yTRL · 2026-04-02
> >
> > The authors have addressed my concerns, and I will raise the score

---

> > > ### Author Response · Authors · 2026-04-03
> > >
> > > We are grateful for your positive assessment and your increased score. The comments provided were very helpful for improving the paper's quality and clarity. We will carefully update the revised version of our manuscript to reflect all of your suggestions.

---

### Official Review · Reviewer_jGrD · 2026-03-13

**Soundness:** 4
**Presentation:** 2
**Significance:** 3
**Originality:** 3
**Overall Recommendation:** 4
**Confidence:** 4

**Summary:**

The paper carries a deep residual learning strategy to model the ocean dynamics and distinct fluid properties. The methodology decomposes the governing dynamics into two components: a physical skeleton and neural flesh. The former is governed by a fast physical system, and the latter by a neural network following a U-Net architecture with a dynamic evaluation layer. The offered method can perform three tasks: simulation, forecasting, and downsampling.

**Compliance With Llm Reviewing Policy:**

Affirmed.

**Final Justification:**

My concerns about the paper are resolved, thus I increased my confidence score. However, there were some discrepancies among the results that the authors put in the rebuttal. I believe there were some mistakes. Overall, my recommendation of this paper is to be accepted.

**Key Questions For Authors:**

- Does OceanDynamicsModel perform worse than FuXi-2.0?

- In line 68, "fitting capability of AI", what is the model architecture that is mentioned as AI?

**Limitations:**

yes

**Strengths And Weaknesses:**

# Strengths:

- Paper brings the deep residual learning to harder to solve problem, modeling ocean dynamics to reduce the computational burden of physical models.

- The offered method outperforms baselines, especially long-term steps.

- Method is evaluated by three important metrics, which show the depth of analysis.

- The method tackles three tasks and shows improvement in all


# Weaknesses:

- The authors made inference analysis in Appendix-G1. This analysis is mainly a comparison between the neural and physical components of the methodology. But comparison with baseline methods in duration for inference and training would deepen the analysis and show the significance of the speed-up gained from the HybridOM method.

- The hybrid method works best when coupled with FuXi-2.0 for forecasting. This questions the need and functionality of the introduced OceanDynamicsModel (ODM). Does the model architecture change when FuXi-2.0 is integrated? This should be explicitly stated in the main paper.

- Authors should provide an explanation for low performance for depther layers.

- Many details in the methodology are omitted and put into the Appendix, making it harder to understand, especially model architecture details in the equations. The notation should be explicit on which function is parameterized with a DL architecture.

- There could be a typo in line 146 while describing equation 4, which states that J_C needs to be F_C.

- Please make the figures SVG or PNG

---

> ### Author Rebuttal · Authors · 2026-03-29
>
> We appreciate reviewer jGrD's insightful comments. We have summarized the reviewer's comments and our responses into the following points:
>
> ### 1. Baseline runtime comparison is missing.
>
> We agree. The experiments in Appendix-G1 are not rigorous. We have now added a unified single-step computation benchmark based on a 0.5-degree global simulation (Table 1 of our original manuscript). The key result is that HybridOM's cost is significantly lower than the SOTA AI-based ocean/weather model:
> | Model | Train time / step | Inference time / step | Peak train memory |
> |---|---:|---:|---:|
> | HybridOM | 782.26 ms | 510.76 ms | 16.29 GB |
> | GraphCast | 1444.36 ms | 1028.07 ms | 45.96 GB |
> | OneForecast | 2338.96 ms | 1508.42 ms | 51.29 GB |
> | NeuralOM | 2162.22 ms | 1273.42 ms | 62.69 GB |
>
> These results support our main efficiency claim: HybridOM is not the absolutely fastest model in every setting, but it is much cheaper than the SOTA data-driven weather/ocean models while preserving the benefits of a differentiable physical skeleton.
>
> Further benchmark details are provided in:
> * [Link to Runtime Benchmark Results](https://anonymous.4open.science/r/ICML2026_rebuttal-6766/response_to_jGrD/runtime_benchmark_results.md)
> * [Link to Efficiency Summary](https://anonymous.4open.science/r/ICML2026_rebuttal-6766/response_to_jGrD/rebuttal_efficiency_summary.md)
>
> ### 2. The role of FuXi-2.0 versus ODM is unclear.
> We agree that this should be stated much more explicitly. `ODM` is the core neural component inside `HybridOM`, and its role is to compensate for the unresolved physics that is not captured by the lightweight dynamical core. By contrast, `FuXi-2.0` is only used in the **forecasting** setting, where it provides the atmospheric boundary forcing required for operational ocean prediction. The two models are not jointly trained. During `HybridOM` training, we use the real atmospheric forcing (`ERA5`) to drive the ocean model. During `FuXi-2.0` training, FuXi is trained as a complete autoregressive atmospheric model with internally matched input/output variables. At forecast time, the parameters of both models are frozen, and they are coupled operationally as shown in Figure 4 of our original manuscript: `FuXi-2.0` supplies atmospheric forcing, while `HybridOM` advances the ocean state under that forcing. Therefore, the gain from coupling `FuXi-2.0` does **not** imply that `ODM` is unnecessary. The coupling is required because operational ocean forecasting needs a physically meaningful atmospheric driver; in our framework, `FuXi-2.0` serves that external boundary-forcing role, whereas `ODM` remains the internal ocean correction module.
>
> ### 3. Low performance in deeper layers needs explanation.
> We agree. Our explanation is that the training objective $\mathcal{L}$ is weighted across variables, not across depth levels within each 3D variable: $\mathcal{L} = \sum_i \lambda_i \|X_i - \hat X_i\|^2$. where $X_i$ and $\hat X_i$ are the predicted and actual values ​​of the i-th variable ($u, v, T, S, \eta$). As a result, upper-ocean layers with stronger variance naturally dominate optimization, while slowly evolving deep layers are harder to improve. This is also partly aligned with physical priorities, because upper-ocean states are more directly tied to air-sea coupling and climate modulation (more important), whereas deep-ocean observations are much sparser (less important).
>
> We now have quantitative depth-sampled evidence. Using four representative depths (`0.49 m`, `55.76 m`, `222.48 m`, `643.57 m`), HybridOM remains best or near-best in most variables. For temperature wMSE:
>
> - lead day `15`: `0.2007 / 0.3212 / 0.2475 / 0.1012` for HybridOM, versus `0.2443 / 0.3629 / 0.2897 / 0.1138` for NeuralOM and `0.2421 / 0.3733 / 0.2887 / 0.1070` for WenHai.
> - lead day `40`: `0.3650 / 0.6740 / 0.5261 / 0.2305` for HybridOM, versus `0.4438 / 0.6996 / 0.5482 / 0.2305` for NeuralOM and `0.5201 / 0.9762 / 0.7791 / 0.2573` for WenHai.
>
> This shows that deeper-layer degradation exists, but HybridOM remains competitive and usually strongest across the sampled depth range.
>
> Extended depth-sampled results are summarized in: [Full_Depth_MSE](https://anonymous.4open.science/r/ICML2026_rebuttal-6766/response_to_jGrD/full_depth_4levels_wmse_summary.md).
>
> ### 4. Method details are too compressed into the appendix.
>
> We agree and will revise our original manuscript. In the revision we will explicitly clarify:
>
> - $J_C$ should be $F_C$ in L146,
> - "fitting capability of AI" refers to strong deep neural network backbone (e.g., `OceanDynamicsModel` in our study)
> - `OceanDynamicsModel` and Learnable Inverse Laplace Operator $\mathcal{N}_{\rm inv}$ are neural-parameterized,
> - expand the methodology section in the revised main text to improve clarity,
> - and the implementation meaning of `K=5`.
>
> ### 5. Figures should be exported as SVG or PNG.
>
> We agree. All final revision figures will be exported as high-resolution `PNG` or `SVG`.

---

> > ### Author Rebuttal · Reviewer_jGrD · 2026-04-03
> >
> > Thank you for the rebuttal. I appreciate the deep work the authors put into responding to my questions. Some of my concerns are answered, specifically, W2 and I believe the authors would update their main manuscript based on their responses for W4 and W5. The remaining questions are as follows;
> > - W1: The results shown in the table for the rebuttal and the results shown in the link are not matching. For example, for HybridOM, the training and inference times are exactly 782.26 ms and 510.76 ms. But, for the training and inference times of OneForecast, GraphCast, and NeuralOM methods, the results shown in the rebuttal are much higher than the table that is located in the link. This is confusing. I would appreciate it if the authors could provide a reason for this discrepancy.
> > - W2: Can authors provide this comparison that is listed in bullet points as a table? This will help me to understand the pattern of errors as we go deeper.
> >
> > Currently, I would like to keep my score and confidence the same.
> >
> > Thank you

---

> > > ### Author Response · Authors · 2026-04-03
> > >
> > > We appreciate the reviewer's positive feedback on our work and the valuable questions. We promise to these add additional results to the revised version of the manuscript based on your suggestions.
> > >
> > > **Response to the discrepancy in inference times:**
> > >
> > > We sincerely thank the reviewer for pointing out this discrepancy. After a detailed check, we found that the inference times reported in our main rebuttal text are incorrect. During the rebuttal period, we had to process and organize a large amount of data. Unfortunately, we made a clerical error: we accidentally copied the incorrect inference time results (based on the same method in Appendix-G1 of our original manuscript) for NeuralOM, GraphCast, and OneForecast into the main rebuttal table. We sincerely apologize for this mistake and any confusion it may have caused.
> > >
> > > Please refer to the correct results provided in our responses to Reviewers **aTzR** and **Ksbq**, which are also available at this anonymous link:
> > > [Efficiency](https://anonymous.4open.science/r/ICML2026_rebuttal-6766/response_to_jGrD/rebuttal_efficiency_summary.md)
> > >
> > > For your convenience, we have included the corrected table below. We will make sure to update the table with these correct results in the revised version of our paper. Overall, the conclusion remains unchanged. Due to the existence of the numerical skeleton, our training speed is significantly faster than that of the baseline models (NeuralOM, OneForecast, GraphCast) compared to inference speed (this is because the physical skeleton has no learnable parameters, so during training, only the gradients of the input variables need to be calculated to be passed back to the neural network).
> > >
> > > | Model | Inference Time | Train Time |
> > > | :--- | :--- | :--- |
> > > | **HybridOM (Ours)** | 510.76 ms | 782.26 ms |
> > > | NeuralOM | 675.11 ms | 2162.22 ms |
> > > | GraphCast | 484.78 ms | 1444.36 ms |
> > > | OneForecast | 689.63 ms | 2338.96 ms |
> > >
> > > **A clearer table for depth-wise wMSE:**
> > >
> > > We agree that presenting this data in a tabular format makes the error patterns much clearer.
> > >
> > > As shown in the table below, HybridOM consistently outperforms or matches the baselines across almost all representative depths. To further highlight the pattern of errors as we go deeper, we have added an "Error Reduction" row, which denotes the relative reduction in wMSE achieved by HybridOM compared to the second-best model in each specific setting. Clearly, error reduction decreases with increasing depth, meaning that our model does indeed have a relatively smaller advantage at deeper levels.
> > >
> > > We believe there are two reasons for this error reduction pattern: 1. Consistent with our original rebuttal, our HybridOM training loss results in relatively less focus on deeper layers. 2. Motion in deeper layers is less affected by rapidly changing atmospheric forcing and originates more from slow changes within the ocean, making it **easier to learn**, thus resulting in similar performance across models.
> > >
> > > Furthermore, as shown in the Tables, this decrease in error reduction primarily occurs below 200m, which is typically the characteristic depth of the mixed layer in the ocean. Below this depth, the ocean is relatively "tranquil." This aligns with our above hypothesis regarding the formation of the error reduction pattern.
> > >
> > > ### Temperature (T) wMSE at Representative Depths
> > >
> > > | Lead Day | Model | 0.49 m | 55.76 m | 222.48 m | 643.57 m |
> > > | :--- | :--- | :--- | :--- | :--- | :--- |
> > > | **Day 15** | **HybridOM** | **0.2007** | **0.3212** | **0.2475** | **0.1012** |
> > > | | NeuralOM | 0.2443 | 0.3629 | 0.2897 | 0.1138 |
> > > | | WenHai | 0.2421 | 0.3733 | 0.2887 | 0.1070 |
> > > | | *Error Reduction (vs best baseline)* | *17.1%* | *11.5%* | *14.3%* | *5.4%* |
> > > | **Day 40** | **HybridOM** | **0.3650** | **0.6740** | **0.5261** | **0.2305** |
> > > | | NeuralOM | 0.4438 | 0.6996 | 0.5482 | 0.2305 |
> > > | | WenHai | 0.5201 | 0.9762 | 0.7791 | 0.2573 |
> > > | | *Error Reduction (vs best baseline)* | *17.8%* | *3.7%* | *4.0%* | *0.0%* |
> > >
> > > ### Zonal Velocity (U) wMSE at Representative Depths
> > >
> > > | Lead Day | Model | 0.49 m | 55.76 m | 222.48 m | 643.57 m |
> > > | :--- | :--- | :--- | :--- | :--- | :--- |
> > > | **Day 15** | **HybridOM** | **0.0089** | **0.0077** | **0.0047** | **0.0026** |
> > > | | NeuralOM | 0.0107 | 0.0093 | 0.0057 | 0.0030 |
> > > | | WenHai | 0.0140 | 0.0113 | 0.0061 | 0.0032 |
> > > | | *Error Reduction (vs best baseline)* | *17.3%* | *17.3%* | *18.5%* | *16.2%* |
> > > | **Day 40** | **HybridOM** | **0.0166** | **0.0147** | **0.0089** | **0.0049** |
> > > | | NeuralOM | 0.0191 | 0.0169 | 0.0102 | 0.0055 |
> > > | | WenHai | 0.0328 | 0.0277 | 0.0138 | 0.0074 |
> > > | | *Error Reduction (vs best baseline)* | *13.2%* | *12.9%* | *12.4%* | *11.4%* |

---

### Decision · Program_Chairs · 2026-04-30

**Decision:**

Accept (regular)

**Comment:**

This paper has received four reviews and they have unanimously recommended to accept this paper. All four reviews are of good quality and the reviewers have engaged in the discussion, replying and even following up to the authors' responses. Except for some relatively minor issues, the responses seem to have largely addressed the reviewers concerns. The reviews have also highlighted that the paper is well written, and the experimental validation is extensive. My own assessment of the significance is that the paper will be of interest to the ICML community interested in Earth sciences applications.

For these reasons, my recommendation is to accept this paper for publiaction at ICML 2026.

I would kindly encourage the authors to incorporate the feedback from the reviewers and the udpates from the discussion. In particular, there are some concerns about the presentation, which are particularly important to consider for the update. For example, the comments that important details of the description of the method are currently delegated to the appendix.

As an additional piece of feedback from my own reading of the paper, I would encourage the authors to explore the possiblity of _visualising_ the results that are currently presented in tables. In particular, Table 1 contains 13 x 11 = 143 5-digit numbers, which requires a cognitive load to interpret. Table 2 is also hard to interpret. In my opinion, these tables are appropriate, for completeness, in the appendix, but the interpretation would largely improve by presenting these results graphically. Finally, I would consider reviewing the bibliography list to ensure that it renders as desired, for example regarding capitalisation (for example, it currently has "lstm" or "era5").